# Rational structure-guided design of a blood stage malaria vaccine immunogen presenting a single epitope from PfRH5

Thomas E Harrison [1,2,4], Nawsad Alam[1,2,4], Brendan Farrell [1,2,4], Doris Quinkert [1,2], Amelia M Lias [1,2], Lloyd D W King[1,2], Lea K Barfod [1,2], Simon J Draper [1,2], Ivan Campeotto [1,3✉] & Matthew K Higgins [1,2✉]

## Abstract

**There is an urgent need for improved malaria vaccine immunogens. Invasion of erythrocytes by *Plasmodium falciparum* is essential for its life cycle, preceding symptoms of disease and parasite transmission. Antibodies which target PfRH5 are highly effective at preventing erythrocyte invasion and the most potent growth-inhibitory antibodies bind a single epitope. Here we use structure-guided approaches to design a small synthetic immunogen, RH5-34EM which recapitulates this epitope. Structural biology and biophysics demonstrate that RH5-34EM is correctly folded and binds neutralising monoclonal antibodies with nanomolar affinity. In immunised rats, RH5-34EM induces PfRH5-targeting antibodies that inhibit parasite growth. While PfRH5-specific antibodies were induced at a lower concentration by RH5-34EM than by PfRH5, RH5-34EM induced antibodies that were a thousand-fold more growth-inhibitory as a factor of PfRH5-specific antibody concentration. Finally, we show that priming with RH5-34EM and boosting with PfRH5 achieves the best balance between antibody quality and quantity and induces the most effective growth-inhibitory response. This rationally designed vaccine immunogen is now available for use as part of future malaria vaccines, alone or in combination with other immunogens.**

**Keywords** Malaria; Vaccines; PfRH5; Monoclonal Antibodies; Rational Vaccine Immunogen Design
**Subject Categories** Immunology; Microbiology, Virology & Host Pathogen Interaction; Structural Biology

## Introduction

Malaria continues to cause death and suffering across large parts of the globe, with an estimated 608,000 deaths and 249 million cases in 2021 (World Health Organisation, 2022). The recent recommendation by the World Health Organisation for use of the first two malaria vaccines, Mosquirix (RTS,S/AS01) (Zavala, 2022) and R21/Matrix-M™ (Datoo et al, 2021), are important developments in the battle against this disease. However, additional vaccines that achieve higher efficacy and durability using simpler dosing regimens would be transformative. While RTS,S and R21 target the pre-erythrocytic stage of the life cycle of the causative agent, *Plasmodium falciparum*, the blood stage is also a promising target for intervention as the symptoms of malaria occur after the merozoite form of the parasite emerges from the liver to infect and replicate within erythrocytes (Draper et al, 2018). A vaccine which prevents erythrocyte invasion would prevent both disease and parasite transmission and could stand alone or be combined with a pre-erythrocytic vaccine immunogen.

The most promising candidates for a blood-stage malaria vaccine are components of the PfPCRCR complex, consisting of PfRH5, PfCyRPA, PfRIPR, PfCSS and PfPTRAMP (Farrell et al, 2024; Scally et al, 2022). Each of these is essential for erythrocyte invasion (Baum et al, 2009; Farrell et al, 2024; Scally et al, 2022; Volz et al, 2016) and the complex has recently been shown to bridge the merozoite and erythrocyte membranes (Farrell et al, 2024; Scally et al, 2022). PfRH5 is the best understood component of PfPCRCR and interacts with human membrane protein complexes containing the erythrocyte receptor basigin (Crosnier et al, 2011; Jamwal et al, 2023; Wright et al, 2014). Knockout of PfRH5 (Baum et al, 2009) or antibodies which target either PfRH5 (Alanine et al, 2019; Douglas et al, 2014; Wright et al, 2014) or basigin (Crosnier et al, 2011; Zenonos et al, 2015) can prevent parasite growth.

PfRH5 is already under development as a malaria vaccine. Immunisation of Aotus with PfRH5 prevents the development of

[1] Department of Biochemistry, Dorothy Crowfoot Hodgkin Building, University of Oxford, South Parks Rd, Oxford OX1 3QU, UK. [2] Kavli Institute for Nanoscience Discovery, Dorothy Crowfoot Hodgkin Building, University of Oxford, South Parks Rd, Oxford OX1 3QU, UK. [3] Present address: School of Biosciences, Division of Microbiology, Brewing and Biotechnology, University of Nottingham, Sutton Bonnington Campus, Sutton Bonington LE12 5RD, UK. [4] These authors contributed equally: Thomas E Harrison, Nawsad Alam, Brendan Farrell. ✉E-mail: Ivan.Campeotto@nottingham.ac.uk; matthew.higgins@bioch.ox.ac.uk

symptoms when animals are challenged with *Plasmodium falciparum* (Douglas et al, 2015). Vaccination of human volunteers with PfRH5 reduces the parasite multiplication rate on parasite challenge (Minassian et al, 2021), although volunteers were not protected from reaching a level of parasitaemia which matched the treatment threshold. This lack of complete protection in humans was attributed to the induction of an insufficient concentration of neutralising antibodies. Indeed, the large number of parasites in the blood, coupled with the brief time window during which they are exposed before they invade another erythrocyte, provide major challenges. The outcomes of current studies therefore show the promise of PfRH5 as a vaccine candidate, while also highlighting the need to use structure-guided, rational approaches to design improved PfRH5-based vaccine immunogens. These should either increase the overall quantity of PfRH5-targeting antibodies or increase the quality of the antibody response by more specifically eliciting just the most neutralising antibodies. This latter strategy is investigated here.

Structure-guided vaccine design (Castro et al, 2022; Sesterhenn et al, 2018) can be used to improve the quality of an immune response by specifically grafting the subdominant epitope for a neutralising monoclonal antibody onto a small scaffold protein (Azoitei et al, 2011; Correia et al, 2014; Schoeder et al, 2021; Sesterhenn et al, 2020). A variety of approaches, including side chain grafting (McLellan et al, 2011; Schoeder et al, 2022), backbone grafting (Azoitei et al, 2011) and, more recently, building novel protein topologies to support a specific epitope (Sesterhenn et al, 2020), can be used to produce such artificial immunogens in which the epitope adopts the correct fold. The designed vaccine immunogen can then specifically elicit a high-quality antibody repertoire (Sesterhenn et al, 2019), with the potential to protect from disease at a lower induced antibody concentration. In the case of PfRH5, our design of a novel immunogen was guided by a crystal structure of PfRH5 in complex with the most effective neutralising monoclonal antibody, which at the time of design was mouse-derived 9AD4 (Wright et al, 2014). The epitope for 9AD4 was grafted onto a small scaffold protein, and Rosetta-based methods were used to redesign the scaffold to ensure correct folding of the epitope, generating an immunogen which elicited a high-quality growth-inhibitory antibody response.

## Results

### Structure-guided design of a focused vaccine immunogen based on PfRH5

The epitope for neutralising antibody 9AD4 is contained entirely within two approximately anti-parallel α-helices which form one side of PfRH5, close to, but not overlapping the basigin binding site. PfRH5 residues 202, 205, 209, 212, 213, 331, 334, 335, 338, 339, 341 and 342 directly contact 9AD4 (Wright et al, 2014) (Fig. 1A; Table EV1). We reasoned that a region of PfRH5 containing this epitope (Fig. 1B) could be recapitulated on a synthetic immunogen built from a molecular architecture consisting of three α-helices, two of which are re-surfaced to present the 9AD4 epitope (Fig. 1C). We identified a three-helical bundle from the *E. coli* ribosome recycling factor with the correct helical topology to allow epitope grafting (PDB 3LHP, chain S) which had previously been used in an

epitope grafting strategy (Correia et al, 2014). While this scaffold protein contains nearly anti-parallel helices, the equivalent helices forming the PfRH5 epitope diverge in their separation across the epitope, suggesting that redesign of the immunogen core would be required to stabilise the conformation of these splayed helices.

We first constructed an in silico model of the immunogen after grafting residues 202–220 and 327–342 from PfRH5 onto the helical bundle (Figs. 1C and EV1). We then used a Rosetta-based design to improve folding of this immunogen. Through this process, the 25 residues which form the epitope surface and make direct contacts with 9AD4 were fixed as invariant (Table EV1), but all other residues were allowed to diversify. This process generated a set of 500 designs, which were scored for their stability, as estimated through the Rosetta score, and their predicted root-mean-squared-deviation from the starting model (Fig. EV1B). Of these, 447 showed broadly equivalent stability and similarity scores. The diversity of the top 100 designs was assessed through evolutionary trace analysis, with one design selected as representative of each branch in the resultant evolutionary tree. These nine designs (Table EV2) showed average pairwise sequence identities from 63 to 84%, with variance found in 53 of the 118 positions on the immunogen.

The nine designs were each expressed in *E. coli* (Fig. EV2A), generating various profiles on a size exclusion column, with some indicative of a correctly folded protein of the predicted size (Fig. 1E). Purified designs were assessed for correct folding by measuring their secondary structure using circular dichroism, with all nine designs showing the expected high α-helical content (Fig. EV2B). To test whether the epitope was correctly folded, we used surface plasmon resonance, flowing the designs over immobilised 9AD4 antibody and measuring the kinetics of binding and dissociation (Figs. 1F and EV3). Each of the designs bound to 9AD4, albeit with differences in binding affinities and kinetics, perhaps representative of different degrees of rigidity of the fold.

We further refined our designs by introducing a disulphide bond to stabilise the correct relative packing of the helices. We selected design 3, which showed a symmetrical size-exclusion chromatography profile, bound 9AD4 with a high affinity and slow off rate and gave the closest predicted root-mean-square deviation to the starting epitope configuration during the design process. We identified two sites in which residues from two neighbouring helices were correctly spaced to allow disulphide formation, CC1 and CC2 (Fig. 1D), and we produced three variants (3A–C), each containing one or two of these disulphides (Fig. EV2). Once again, these were expressed in *E. coli*, generating symmetric profiles on a size-exclusion column (Fig. 1E). Circular dichroism was used to show that they adopt the correct fold (Fig. EV2b). Surface plasmon resonance was then used to test their binding to 9AD4, showing an improvement in the affinity of all three designs when compared with design 3 (Figs. 1F and EV3).

### Human monoclonal antibodies with high potency target the 9AD4 epitope

During the production and testing of the synthetic immunogen, two studies were published which identified growth-neutralising human antibodies which target PfRH5. As these studies identified

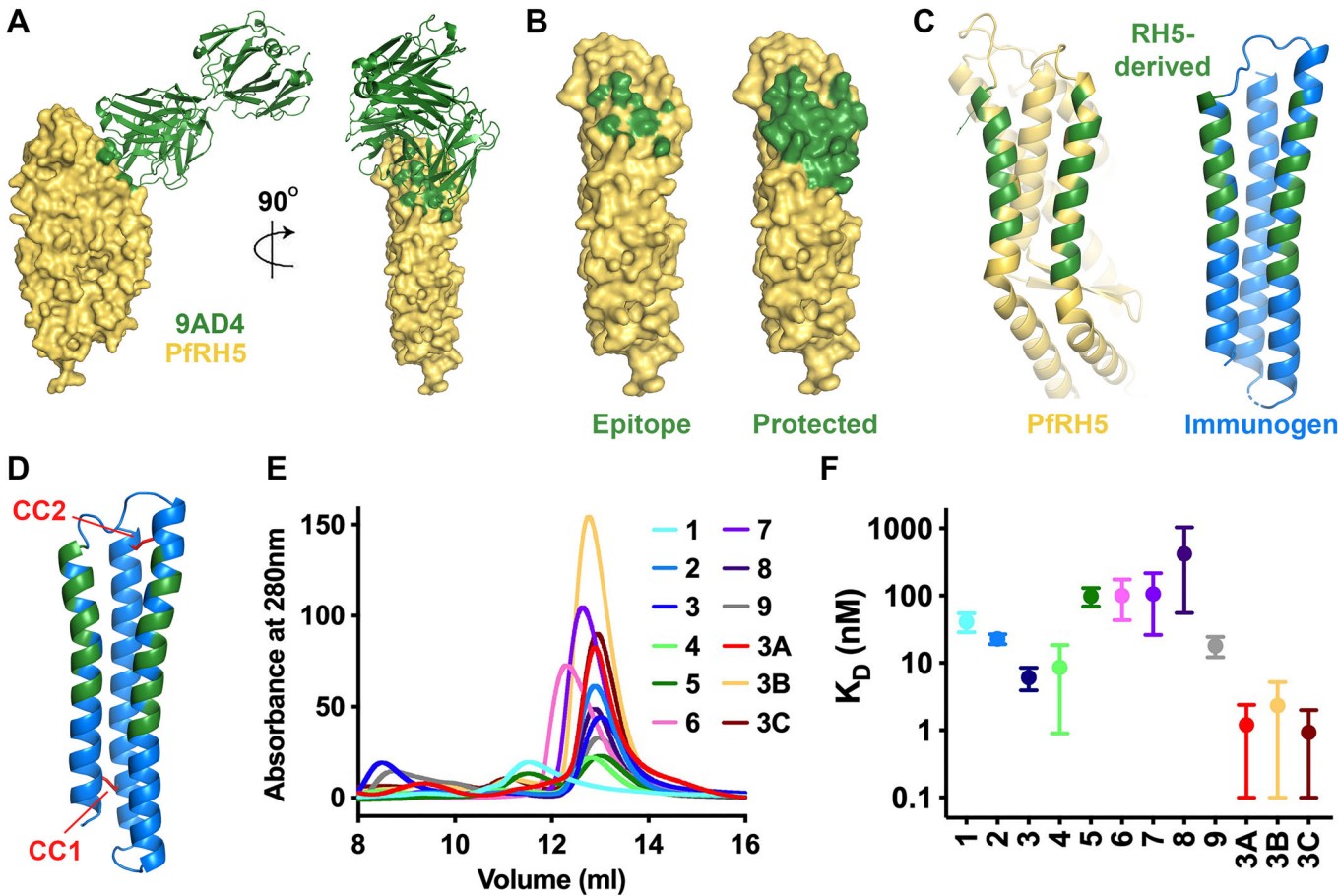

**Figure 1. Design of a synthetic epitope mimic displaying the 9AD4 antibody epitope.**

(A) The structure of PfRH5 (yellow) bound to monoclonal antibody 9AD4 (green, PDB:4UOR) (Wright et al, 2014). (B) Surface representation of PfRH5 in yellow. In the left-hand panel, residues coloured green directly contact 9AD4 while the right-hand panel shows residues which define the broader 9AD4 epitope in green. (C) The left-hand panel shows the structure of PfRH5 (yellow) with the epitope residues coloured green. The right-hand panel shows the designed synthetic immunogen (blue) with the grafted residues in green. (D) The location of two disulphide bonds introduced to stabilise the synthetic immunogen are shown in red and labelled CC1 and CC2. (E) Size-exclusion chromatography traces obtained for the 12 designs (representative from $n = 2$). (F) $K_D$ values obtained from surface plasmon resonance analysis of the binding of the 12 synthetic designs to immobilised antibody 9AD4, shown as mean with error bars representing the 95% confidence interval from a least squares fit (representative from $n = 2$). Source data are available online for this figure.

human antibodies which were effective in growth-inhibition assays at lower concentrations than those required for 9AD4, it was important to determine whether our synthetic immunogen can also mimic the epitope for these antibodies.

The first study isolated and characterised a panel of 17 antibodies (Alanine et al, 2019). The most effective growth-inhibitory antibody from this panel, R5.016, bound to an epitope which overlapped that of 9AD4. We previously determined the structure of the R5.016 bound to PfRH5 (Alanine et al, 2019) and find that 13/15 of the residues on PfRH5 contacted by R5.016 are retained in our epitope mimic designs (Table EV1).

More recently, a panel of 236 monoclonal antibodies was isolated from human volunteers who had been vaccinated as part of a clinical trial (Barrett et al, 2024). Of these, the eight with the lowest $EC_{30}$ for growth inhibition all shared an overlapping binding site on PfRH5 with 9AD4 and R5.016. These included antibody R5.034, which has an $EC_{50}$ 8.3-fold lower than that of R5.016. To determine the degree to which the epitope for R5.034 is

recapitulated in the epitope mimic, we determined the crystal structure of the Fab fragment of R5.034 bound to a version of PfRH5 lacking the flexible N-terminus and the flexible central loop (PfRH5ΔNL) at 2.4 Å resolution (Fig. 2A; Table EV3). We found that R5.034 binds to an epitope which overlaps those of both 9AD4 and R5.016. However, the epitope for R5.034 is more compact than that for R5.016 and is located further from the tip of the PfRH5 diamond. None of these three epitopes include residues found to be polymorphic in PfRH5. The R5.034 epitope is recapitulated in the synthetically designed immunogen with 9/10 of the PfRH5 residues contacted by R5.034 also found in the immunogen (Fig. 2B; Table EV1). These studies therefore support the design of immunogens which recapitulate the epitope for R5.016 and R5.034, as well as that for 9AD4, as a potential route to inducing the most effective PfRH5-targeting growth-inhibitory antibodies following human vaccination. This indicates that our epitope mimic, designed to recapitulate the 9AD4 epitope, may induce R5.016 and R5.034-like antibodies.

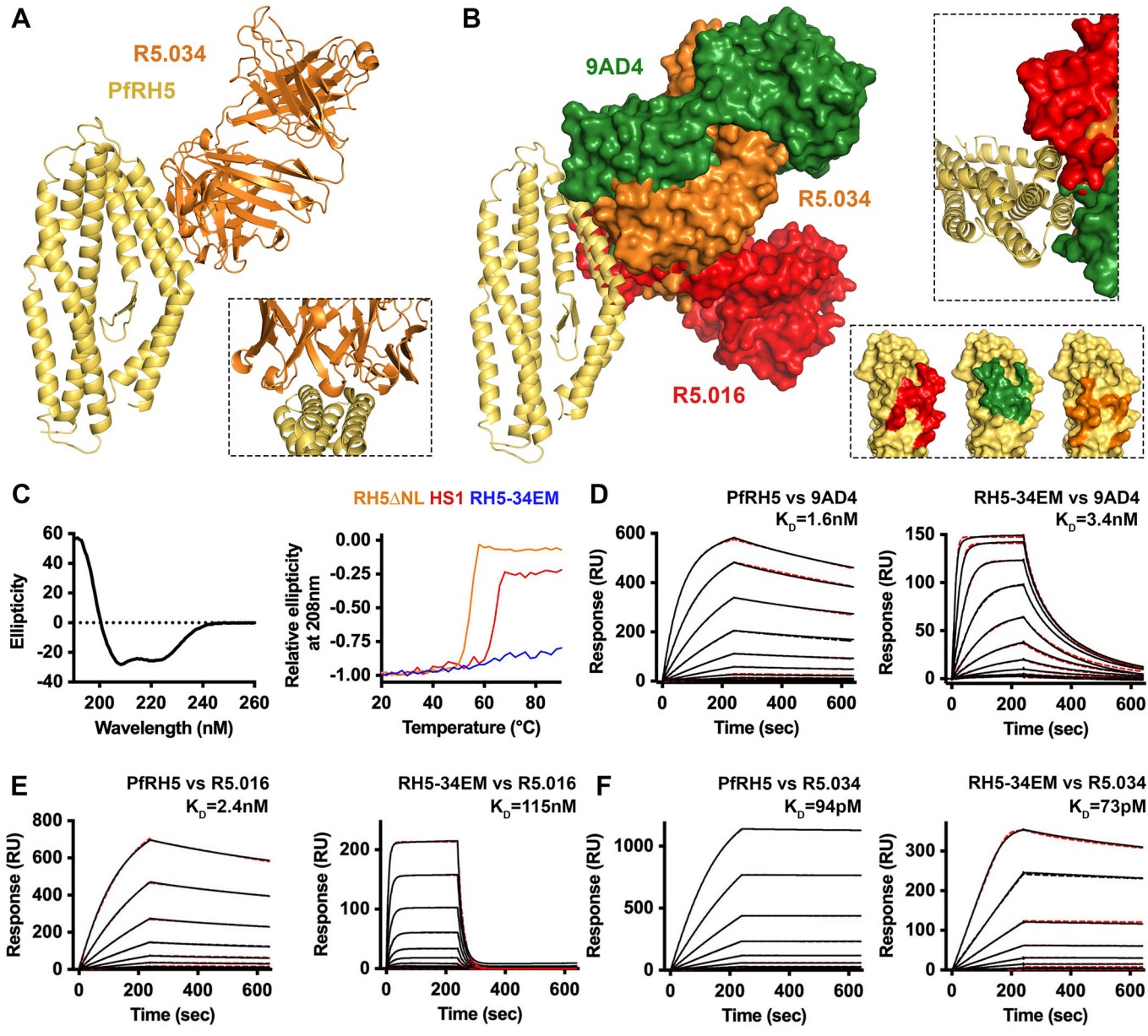

**Figure 2. The epitope mimic binds growth-neutralising antibodies 9AD4, R5.016 and R5.034.**

(A) The structure of PfRH5 (yellow) bound to the Fab fragment of growth-inhibitory monoclonal antibody R5.034 (orange). The inset shows a close-up on the epitope region. (B) An alignment of the structures of PfRH5 in cartoon form (yellow) bound to the Fab fragments of 9AD4 (green), R5.016 (red) and R5.034 (orange) in surface representation. The right-hand panel shows that all three antibodies bind to the two helices recapitulated in the epitope mimic, while approaching it from different angles. (C) The left-hand panels show a circular dichroism trace of RH5-34EM. The right-hand panel shows the effect of increasing temperature on the ellipticity at 208 nm of RH5ΔNL (Wright et al, 2014) (orange), thermally stabilised RH5_HS1 (Campeotto et al, 2017) (red) and RH5-34EM (blue), showing temperature stability. Surface plasmon resonance traces of PfRH5 and RH5-34EM binding to (D) 9AD4; (E) R5.016; (F) R5.034, with the data shown as black lines and fits to a 1-to-1 binding model shown as red dashed lines and representative of $n = 2$. Source data are available online for this figure.

## Structural and biophysical assessment of the epitope mimic

We next used structural and biophysical studies to test the hypothesis that the epitope mimic replicates the epitope for 9AD4, as well as those for human neutralising antibodies R5.016 and R5.034. We selected design 3A to progress to detailed characterisation and provisionally called this RH5-34EM,

according to our hypothesis that it is an excellent mimic of the R5.034 epitope. We conducted circular dichroism analysis at increasing temperatures and show that secondary structure content is retained up to 90 °C (Fig. 2C), compared with a melting temperature of ~65 °C for our previous thermally stabilised PfRH5_HS1 design (Campeotto et al, 2017), now also called RH5.2 (King et al, 2024). We then used surface plasmon resonance (SPR) to measure the binding kinetics of 9AD4, R5.016 and R5.034

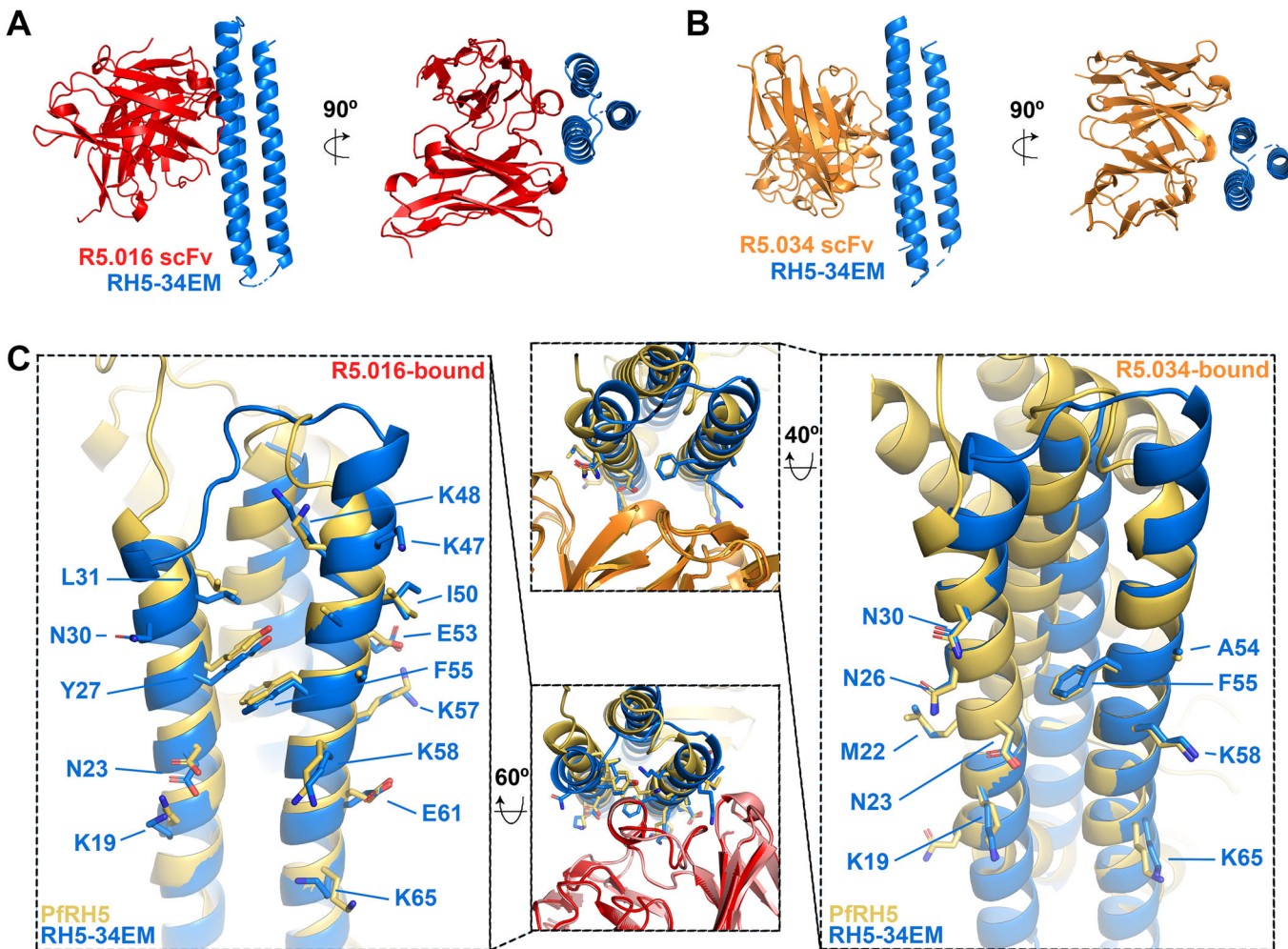

**Figure 3. Structures of RH5-34EM bound to antibodies R5.016 and R5.034.**

Crystal structures of RH5-34EM (blue) bound to the scFv fragment of antibody (**A**) R5.016 (red) and (**B**) R5.034 (orange) viewed from two directions. (**C**) Alignments of antibody-bound structures of RH5-34EM and PfRH5. The left-hand panel shows an alignment of the R5.016-bound forms of RH5-34EM (blue) and PfRH5 (yellow). Side chains that contact R5.016 are shown as sticks and are labelled according to the numbering of RH5-34EM. The right-hand panel shows an alignment of the R5.034-bound forms of RH5-34EM (blue) and PfRH5 (yellow). Side chains that contact R5.034 are shown as sticks and are labelled according to the numbering of RH5-34EM. The central panels showing the same alignments, viewed from different angles and showing bound antibodies.

for both RH5-34EM and PfRH5. RH5-34EM bound to 9AD4 with an affinity of 3.4 nM compared with 1.6 nM for PfRH5, albeit with a faster on-rate and faster-off rate (Fig. 2D; Table EV4). R5.016 bound with lower affinity to RH5-34EM than PfRH5, although still in the nanomolar range (115 nM vs 2.4 nM) (Fig. 2E; Table EV4). However, R5.034 showed a very similar high affinity and binding kinetics for both RH5-34EM and PfRH5 (73 pM vs 94 pM) (Fig. 2F; Table EV4). Therefore, RH5-34EM effectively mimics the epitope of the most effective growth-inhibitory human antibody, R5.034 and substantially mimics that of R5.016.

We next determined crystal structures of the RH5-34EM epitope mimic bound to scFv fragments of human monoclonal antibodies R5.016 (at 1.63 Å resolution) and R5.034 (at 1.75 Å resolution) (Fig. 3A,B; Table EV3). Comparison of these structures with those of PfRH5 bound to R5.016 and R5.034 showed significant similarity, both in the fold of the epitope mimic and in the interactions that it makes with antibodies. Of the 15 interactions

formed between PfRH5 and R5.016, 13 were retained with the epitope mimic. In the case of R5.034, of the 10 interactions made with PfRH5, 9 were also observed with the epitope mimic (Fig. 3C; Table EV1). These structural alignments also confirmed that RH5-34EM more correctly recapitulates the epitope for R5.034 than that for R5.016, with R5.034 binding closer to the centre of the RH5-34EM immunogen, while R5.016 binds more towards the tip of the immunogen, where the structural similarity to PfRH5 starts to diverge (Fig. 3C). These structural and biophysical studies confirm that RH5-34EM is a small, highly stable synthetic immunogen which recapitulates the epitopes for neutralising antibodies R5.034, R5.016 and 9AD4 on a small stable scaffold.

## RH5-34EM generates a high-quality immune response

We next aimed to assess the immunogenicity of RH5-34EM in comparison to that of PfRH5. We immunised cohorts of six rats

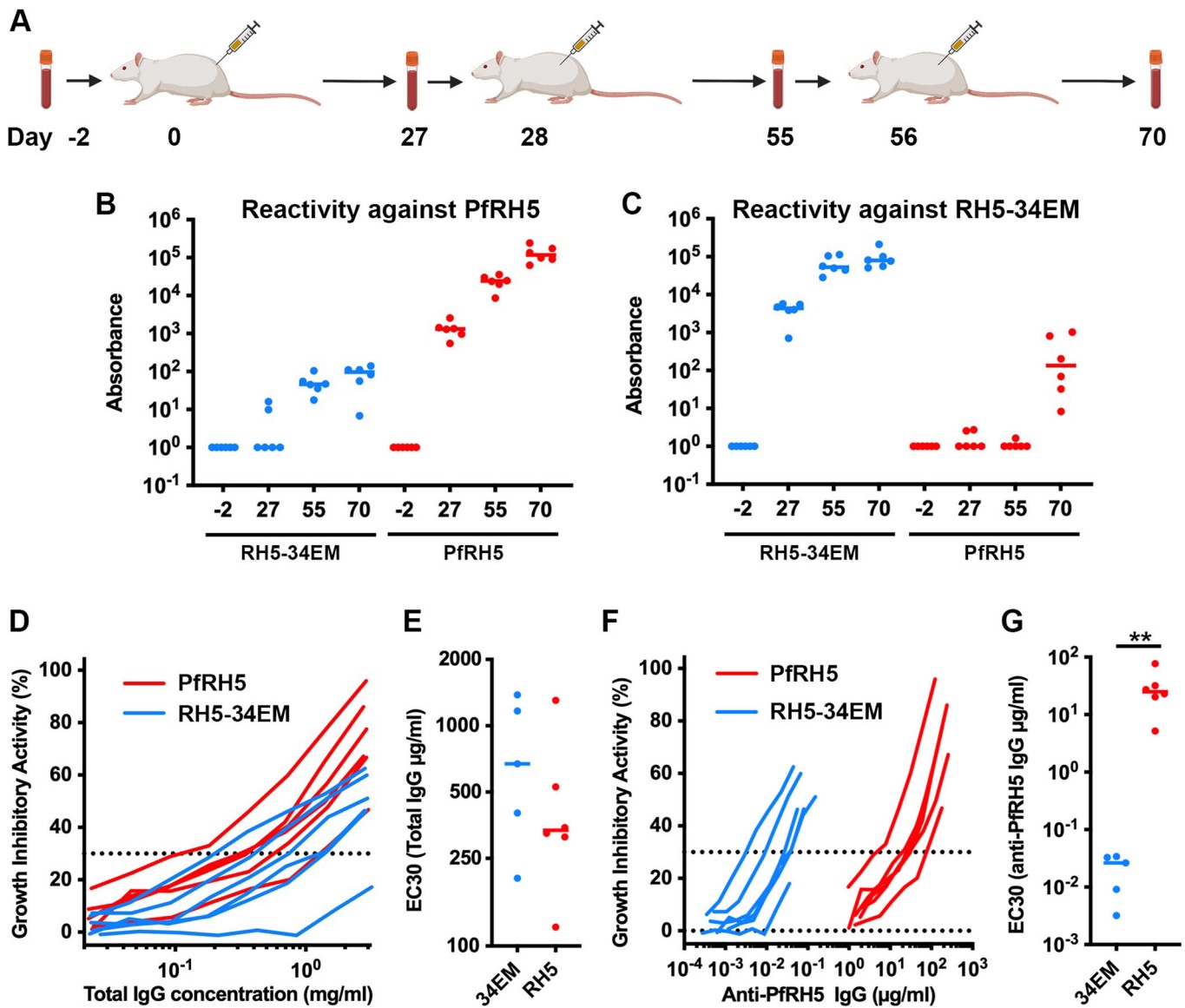

**Figure 4. Comparison of antibodies induced through immunisation with RH5-34EM and PfRH5.**

(A) Scheme for rat immunisations with RH5-34EM and PfRH5. Blood samples were taken on days −2, 27, 55 and 70 and immunisations were conducted on days 0, 28 and 56. (B) Sera from rats immunised with RH5-34EM (blue) and PfRH5 (red) were assessed for their binding to immobilised PfRH5 by ELISA. (C) Sera from for rats immunised with RH5-34EM (blue) and PfRH5 (red) were assessed for their binding to immobilised RH5-34EM by ELISA. (D) Growth-inhibitory activity of different concentrations of total IgG purified from sera raised by immunisation of rats with RH5-34EM (blue) and PfRH5 (red). (E) The $EC_{30}$ of IgG from (D). (F) Growth-inhibitory activity of different concentrations of IgG purified from sera raised by immunisation of rats with RH5-34EM (blue) and PfRH5 (red), calibrated for the amount of PfRH5-specific IgG. (G) The $EC_{30}$ of IgG from (F). In each cohort, we immunised six rats and analysed them individually. Statistical significance determined using a two-tailed Mann–Whitney test (**$P < 0.005$, for 34EM vs RH5, $P = 0.0043$). Source data are available online for this figure.

with either three doses of RH5-34EM or three doses of the full-length PfRH5 immunogen, PfRH5.1 (Jin et al, 2018), both formulated with the Matrix-M™ adjuvant (Stertman et al, 2023) (Fig. 4A). As small immunogens are likely to induce lower antibody responses, we conjugated RH5-34EM, through a spy tag at the N-terminus, to a virus-like particle consisting of the hepatitis B surface protein (HBsAg) fused to spy-catcher (Marini et al, 2019). We compared this with soluble PfRH5.1 formulated with the adjuvant Matrix-M™, which is currently the lead PfRH5-based clinical candidate. This allows the outcomes of this study to be

compared directly with previous studies in both animal models (Douglas et al, 2015) and humans (Minassian et al, 2021; Silk et al, 2024). Fourteen days after the final dose, sera were harvested and were analysed for antibody responses against both RH5-34EM and PfRH5 using ELISA.

We first measured the amount of PfRH5-specific IgG induced by each immunogen. Immunisation with RH5-34EM required two injections to induce PfRH5-specific antibodies, with little increase in titre after a third injection (Fig. 4B). However, immunisation with PfRH5 resulted in around a thousand-fold higher titre of

PfRH5-specific antibodies after three immunisations. When studying RH5-34EM-specific antibodies, we found that immunisation with RH5-34EM generated high titres after just one immunisation (Fig. 4C). In contrast, immunisation with PfRH5 required three immunisations to induce RH5-34EM-specific antibodies at a titre approximately a 1000-fold lower than those induced through immunisation with RH5-34EM (Fig. 4C).

We next assessed the quality of the induced antibodies in a growth-inhibition assay. This assay studies the reduction in in vitro growth of *Plasmodium falciparum* parasites in human red blood cells in the presence of antibodies. In this case, we first purified IgG from the sera, allowing us to reach antibody concentrations sufficient to observe growth-inhibitory activity and removing background effects. We first studied the growth-inhibitory activity of total IgG, as this is most representative of sera and most relevant to the protective response in a vaccinated individual. Total IgG were purified from rat sera on day 70 and the growth-inhibitory activity (GIA) was assessed for a dilution series of these antibodies (Fig. 4D,E). The antibody concentration required for 30% GIA ($EC_{30}$) was around twofold lower for sera from PfRH5-immunised rats than for sera from those immunised with RH5-34EM, albeit this difference was not statistically significant.

In addition, to assess the quality of the antibody responses, we also determined the growth-inhibitory activity after calibration for the concentration of PfRH5-specific antibodies, rather than total IgG. This allowed us to assess the growth-inhibitory potential as a factor of only PfRH5-specific antibodies, revealing whether PfRH5-specific antibodies induced by RH5-34EM were more effective than those induced by PfRH5. ELISA was used to measure the concentration of PfRH5-specific antibodies in both sera, allowing conversion of these data to reveal the effect of PfRH5-specific IgG on growth (Fig. 4F,G). In this case, the PfRH5-specific antibodies from RH5-34EM-immunised rats showed an $EC_{30}$ approximately one thousand-fold lower than those from PfRH5-immunised rats.

Therefore, the growth-inhibitory quality of the PfRH5-specific antibodies induced through immunisation with RH5-34EM was around one thousand-fold greater that of the PfRH5-specific antibodies induced through immunisation with PfRH5. However, when considered in terms of the growth-inhibitory activity as a factor of total IgG, immunisation with PfRH5 is around twofold more effective, suggesting that PfRH5 immunisation generates more antibodies targeting multiple neutralising epitopes on PfRH5.

## Priming with a focused immunogen followed by a PfRH5 boost generates the most growth-inhibitory antibody response

As immunisation with RH5-34EM generated a high-quality, low-quantity response, while immunisation with PfRH5 generated a lower-quality, high-quantity response, we next aimed to determine whether different prime-boost regimens, in which RH5-34EM and PfRH5 are provided in different orders, could generate a high-quality, high-quantity response. We also assessed whether it is more effective to prime with the focused RH5-34EM immunogen, followed by boost with a broader PfRH5-targeting response, or to reverse the sequence, priming for a broad PfRH5-targeting response and then boosting the epitope-specific response using the more focused RH5-34EM immunogen. We therefore immunised rats, using the same timings and immunogens as Fig. 4A, but

including each of the six different remaining possible orders for the successive doses of PfRH5 and RH5-34EM immunogens (Figs. 5, EV4 and EV5).

ELISA measurements were used to assess the PfRH5-specific and RH5-34EM-specific responses resulting from these immunisation regimens and were largely as predicted. ELISA reactivity against RH5-34EM scaled with the number of doses of RH5-34EM, with two or three doses of RH5-34EM generating equivalent reactivity, one dose of RH5-34EM generating an intermediate reactivity and use of only PfRH5 doses generating the lowest reactivity (Fig. 5A). Apart from the PfRH5–PfRH5–RH5-34EM regimen, the order in which doses were provided did not affect overall reactivity against RH5-34EM. Indeed, the PfRH5–PfRH5–RH5-34EM regime was inconsistent with the other regimes, as the animals didn't respond to the PfRH5 prime in the same way as in other cohorts, leading to questions about whether this prime dose was administered correctly.

When assessing PfRH5-reactivity, we also observed the number of PfRH5 doses to scale with the ELISA reactivity (Fig. 5B). Here, similar reactivity resulted from two or three doses of PfRH5, provided in any order. One dose of PfRH5, provided at any of the three immunisation time points, resulted in an intermediate PfRH5-reactivity, while the lowest PfRH5-reactivity was generated using three doses of RH5-34EM. Again, no significant difference was observed depending on the point at which PfRH5 was provided in the immunisation sequence. When immunising with RH5-34EM, the ELISA response reached ~$10^5$ against RH5-34EM when compared with ~$10^2$ against PfRH5. While these ELISA assays are not directly comparable, this nevertheless suggests that most antibodies elicited by RH5-34EM target the scaffold. Despite this, RH5-34EM does elicit PfRH5-reactive antibodies.

We next assessed these samples in the growth-inhibition assay as above, measuring the functional activity of total purified IgG and PfRH5-specific IgG. When studying the quality of the response by considering PfRH5-specific IgG, the data followed the pattern expected (Fig. 5C). Three doses of RH5-34EM generated PfRH5-specific IgG with the lowest $EC_{30}$. Next most effective was two doses of RH5-34EM and one dose of PfRH5. These IgG had a significantly ($P = 0.0303$ to $P = 0.0043$) higher $EC_{30}$ than those induced through three RH5-34EM doses. In addition, the position within the immunisation sequence of the PfRH5 dose had a significant effect, with priming with RH5-34EM and boosting with PfRH5 generating a higher quality response than using the PfRH5 dose at an earlier time point. Therefore, priming to obtain a focused response against a specific epitope, followed by boosting to broaden out the response, is more effective than incorporating the focused immunogen as the second or third doses. Finally, two or three doses of PfRH5 generated the highest $EC_{30}$, with no significant difference between these four cohorts. Therefore, while RH5-34EM alone generates the highest quality PfRH5-specific response, two doses of RH5-34EM followed by one of PfRH5 induces PfRH5-specific antibodies with a fourfold weaker $EC_{30}$ and a more effective growth-inhibitory response is induced when the focused immunogen is used early in the dosing regimen. A similar pattern was observed when considering $EC_{50}$, although this required omitting some data points which didn't reach this degree of inhibition at the maximum IgG concentration possible (Fig. EV5).

Finally, we assessed the growth-inhibitory activity from total IgG (Fig. 5D). Here, rat-to-rat variation meant that the differences

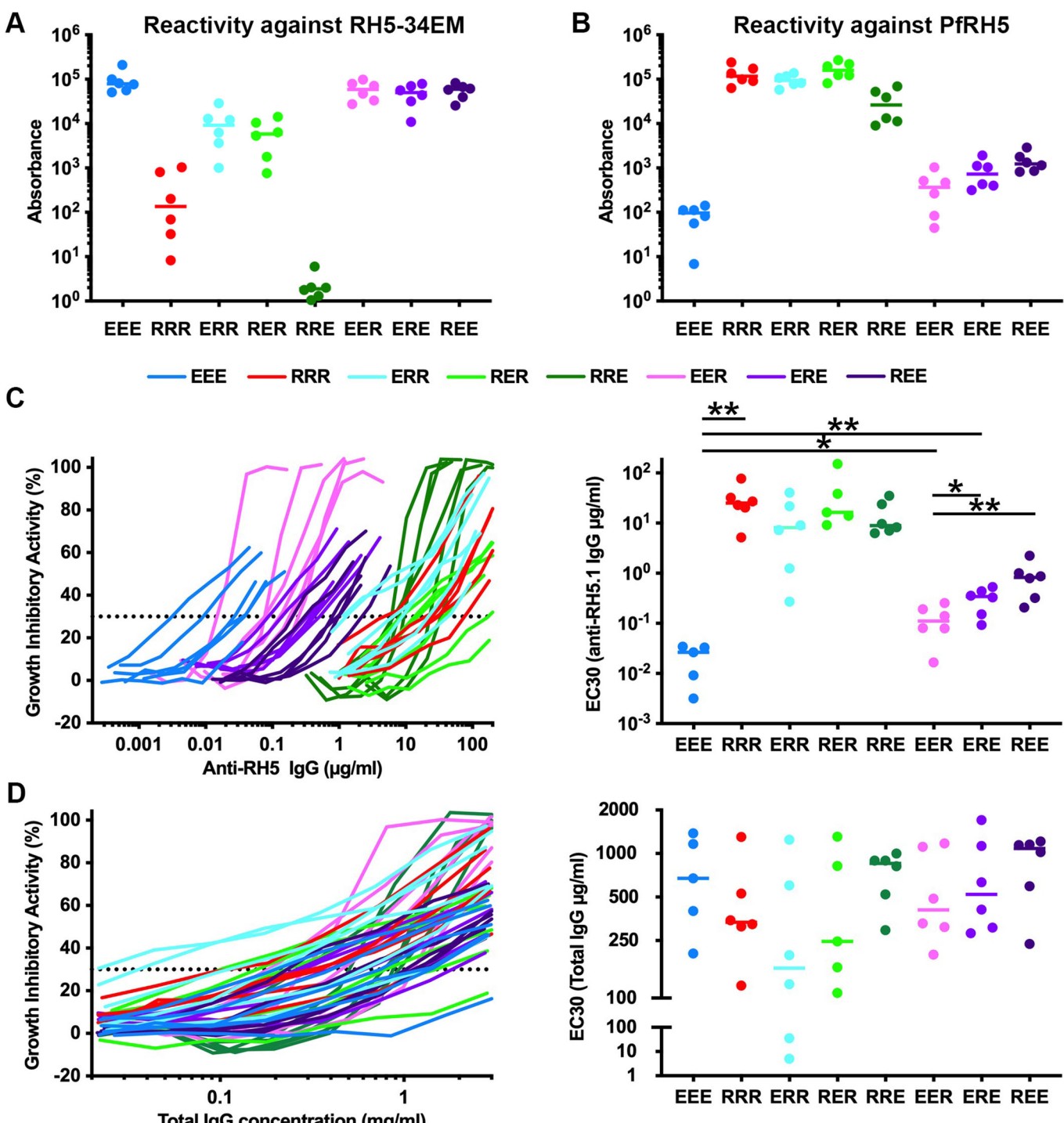

**Figure 5. Comparison of different prime-boost regimens for RH5-34EM and PfRH5.**

RH5-34EM (E) and PfRH5 (R) were used in different prime-boost regimens, using the same dosing schedule as Fig. 4A. The three-letter code used in this figure gives the dosing regimen: for example RER is a PfRH5 dose on day 0, a RH5-34EM dose on day 28 and a PfRH5 dose on day 56. The data for RRR and EEE is the same as that shown in Fig. 4. Sera from immunised rats were assessed for binding to (**A**) RH5-34EM and (**B**) PfRH5 by ELISA. (**C**) Growth-inhibitory activity (left) and its $EC_{30}$ (right) for PfRH5-specific IgG. Statistical significance determined using a two-tailed Mann–Whitney test corrected for multiple comparisons (* indicates <0.05 and ** indicates <0.005; for EEE vs RRR, $P = 0.0043$; for EEE vs EER, $P = 0.0303$; for EEE vs ERE, $P = 0.0079$; for EER vs ERE, $P = 0.0411$; for EER vs REE, $P = 0.0043$). (**D**) Growth-inhibitory activity (left) and its $EC_{30}$ (right) for total IgG. In each cohort, we immunised six rats and analysed individually. Source data are available online for this figure.

between these groups were not statistically significant. However, the lowest mean $EC_{30}$ resulted from one dose of RH5-34EM followed by two doses of PfRH5. The next lowest involved a dose of PfRH5 followed by RH5-34EM followed by PfRH5. These two regimens both resulted in approximately twofold lower $EC_{30}$ for growth inhibition than three doses of PfRH5. Therefore, the leading dosing regimen in terms of growth-inhibitory activity from total IgG is one dose of a focused RH5-34EM immunogen, followed by two of the full PfRH5 immunogen.

## Discussion

Structure-based antigen design aims to rationally design vaccines by structurally characterising the most effective monoclonal antibodies that result from vaccination or infection and designing new generations of vaccine immunogens to specifically recapitulate their epitopes (Burton, 2002). The aim is to specifically elicit the most protective neutralising antibodies, without also inducing ineffective antibodies or antibodies which might interfere with neutralising antibody function. In its most extreme form, this approach involves generating highly focused immunogens which specifically present single epitopes for the most effective neutralising antibodies. It is this approach which we deploy here for PfRH5.

When we started this study, we were guided by structures of PfRH5 in complex with mouse-derived monoclonal antibodies. The most effective growth-inhibitory antibody available at the time, 9AD4 (Douglas et al, 2014), bound to an epitope consisting of two α-helices on the side of PfRH5 (Wright et al, 2014). We therefore decided to design a synthetic immunogen presenting this most neutralising epitope. Subsequent studies have supported this decision. In a panel of 18 monoclonal antibodies from human volunteers immunised with PfRH5, the most effective antibody, R5.016 bound to the same epitope as 9AD4 (Alanine et al, 2019). In a recent study of 236 antibodies isolated from vaccinated volunteers, the most effective growth-inhibitory antibodies, including the new most potent PfRH5-targeting antibody, R5.034, all target the same epitope as 9AD4 and R5.016 (Barrett et al, 2024) and we present the structure of PfRH5 bound to R5.034 here (Fig. 2A). Given that these antibodies bind epitopes of PfRH5 which do not overlap the basigin binding site, their mechanism of action was previously unclear. However, we recently showed that basigin is exclusively part of transmembrane complexes on the erythrocyte surface and that 9AD4 and R5.016 both sterically prevent PfRH5 from binding to basigin in this context (Jamwal et al, 2023). Therefore, the 9AD4/R5.016/R5.034 epitope remains that most likely to induce a high-quality growth-inhibitory antibody response.

Production of our focused immunogen involved grafting the 9AD4/R5.016/R5.034 epitope onto a small scaffold protein and redesigning the resultant molecule to ensure that it folds correctly. This was successful, with structural studies showing that both R5.016 and R5.034 bind to the synthetic immunogen with the expected binding mode. Surface plasmon resonance analysis confirmed the formation of the correct epitope, with all three antibodies binding with at least nanomolar affinity, albeit showing that the immunogen most effectively recapitulated the R5.034 epitope. We therefore achieved our goal of producing a focused immunogen, RH5-34EM, which specifically presents a single epitope from PfRH5.

Immunisation studies with this focused immunogen were conducted in rats and the outcome was analysed using ELISA and growth-inhibition assays to assess the quantity and quality of induced antibodies. These data showed that immunisation with three doses of RH5-34EM generated an approximately one thousand-fold lower PfRH5-specific antibody titre than the same number of doses of PfRH5. Indeed, this difference may have been even greater had we used PfRH5.1 conjugated to a virus-like particle as the comparator, instead of the lead clinical vaccine, soluble PfRH5.1. However, immunisation with RH5-34EM-induced antibodies that were around one thousand-fold more potent than those induced by PfRH5, as determined by the $EC_{30}$ for growth-inhibitory activity due to PfRH5-specific antibodies. A similar pattern was seen for RSV-targeting antibodies, with a highly focused single-epitope vaccine inducing a lower quantity, higher quality response (Sesterhenn et al, 2019; Sesterhenn et al, 2020). Studies with the malaria vaccine immunogen PfCSP provide a possible mechanism through which epitope-specific responses might be limited, with antibodies induced by an early immunisation dose binding to, and masking, the epitope when presented in subsequent vaccine doses (McNamara et al, 2020). Indeed, in our study, PfRH5-specific responses induced by RH5-34EM were not appreciably increased by a third immunisation. Future studies will show whether this is a general effect for highly focused single-epitope immunogens.

The availability of the highly focused RH5-34EM immunogen next allowed us to determine whether it is more effective to start by inducing an epitope-specific response, through RH5-34EM immunisation, followed by a broader PfRH5 boost, or whether it is more effective to start with a broad PfRH5-based response which is then focused onto a single epitope through RH5-34EM immunisation. Previous studies using highly focused, single-epitope-containing immunogens have used the latter approach, immunising with a larger immunogen and then attempting to boost the subdominant, protective antibodies using an epitope-specific immunogen (Sesterhenn et al, 2019). In our case, we also assessed the effect of immunisation with a focused immunogen to elicit antibodies against the 9AD4/R5.016/R5.034 epitope, followed by a boost to generate a broader PfRH5 response. This regimen gave the best balance of quality and quantity of immune response, with the lowest mean $EC_{30}$ of growth inhibition from total IgG, achieved at 375-fold less PfRH5-specific IgG than that induced by PfRH5 alone. Therefore, starting with a highly focused immune response and then allowing this response to both boost and broaden appears to be an effective strategy.

While a RH5-34EM prime followed by two PfRH5 boosts gives a better growth-inhibitory response than three doses of PfRH5, the need for two distinct immunogens, provided in the correct sequence, would bring challenges for a human vaccination strategy in malaria-endemic regions. Two approaches will therefore be used to improve upon RH5-34EM. The first will aim to improve epitope-specific immunogenicity while reducing immune responses to the immunogen backbone, thereby increasing the quantity of the PfRH5-specific response while not reducing its quality. This might be achieved by multimerization of RH5-34EM such that surfaces which do not contribute epitopes are concealed by multimerization interfaces, or by altering presentation of RH5-34EM on VLP

scaffolds to increase the presentation of the epitopes and decrease presentation of the scaffold. A second approach will be to generate vaccine immunogens which contain a larger antigenic surface than RH5-34EM. Our studies of antibodies targeting the PfRH5-binding partner PfCyRPA revealed that antibodies which bind to neighbouring epitopes can function synergistically, through the formation of lateral antibody-antibody interactions (Ragotte et al, 2022). Antigens which contain broader epitope regions might therefore be more effective than multiple highly focused immunogens. In the case of PfRH5, there are two major epitopes for neutralising antibodies in addition to the one targeted here (Alanine et al, 2019; Wang et al, 2024). These epitopes lie around the top surface of PfRH5, providing a continuous antigenic surface (Alanine et al, 2019; Wang et al, 2024). Future vaccine design strategies will therefore focus on generating synthetic immunogens which recapitulate this entire surface, reducing the chance that antibody masking will decrease immunogenicity and increasing the likelihood of inducing synergistic antibody repertoires.

PfRH5 remains the most promising vaccine candidate with which to target the blood stage of malaria and continues to progress in clinical trials. PfRH5 is also the first malaria vaccine candidate to benefit from structure-guided vaccine design. A first-generation structure-guided design for PfRH5 involved truncation of a flexible N-terminus and flexible loop, as well as thermal stabilisation (PfRH5_HS1) (Campeotto et al, 2017). This immunogen, now known as RH5.2, is in clinical trial (King et al, 2024). We now add to this an additional immunogen, RH5-34EM, which recapitulates the epitope for the most neutralising monoclonal antibodies and show that this induces a high-quality immune response. These immunogens can now be tested to find the most effective combination for the prevention of malaria.

## Limitations of the study

The primary limitation of this study results from our use of a model for assessing vaccine efficacy in which we immunise rats, purify IgG and assess the growth-inhibitory effect of the purified IgG in an in vitro assay. This model is used as a standard in the field due to the lack of a parasite challenge model which can be deployed at high throughput to screen multiple conditions. The immunisation of six rats with each vaccine provides sufficient IgG to assess the growth-inhibitory activity of antibodies from each individual animal, allowing animal-to-animal variation to be assessed. The growth-inhibition model has disadvantages, such as the requirement to use purified IgG at sufficient concentrations, rather than studying sera. In this study, this led us to determine EC30 rather than EC50, as insufficient IgG was elicited to reach EC50 in some cases. The use of a rodent model also brings the risk that the antibodies elicited will not be the same as those elicited in humans. Nevertheless, comparative analysis of growth-inhibitory activity with efficacy in aotus (Douglas et al, 2015) and human challenge studies (Minassian et al, 2021; Silk et al, 2024) suggests that the growth-inhibition model is indicative of efficacy, giving us confidence that the outcomes of this study will be transferrable to vaccine challenge trials.

# Methods

### Reagents and tools table

| Reagent/resource | Reference or source | Identifier or catalogue number |
| --- | --- | --- |
| **Experimental models** | | |
| Shuffle T7 | New England Biolabs | C3026J |
| Expi293F™ cells | Thermo Fisher | A14527 |
| FreeStyle™ 293-F cells | Thermo Fisher | R79007 |
| S2 cell line | ExpreS²ion Biotechnologies | N/A |
| Female Wistar IGS rats | Noble Life Sciences | N/A |
| *Plasmodium falciparum* 3D7 strain | https://www.beiresources.org/Catalogue/BEIParasiticProtozoa/MRA-102.aspx | N/A |
| **Recombinant DNA** | | |
| pEt15b vector | Novagen | 69661 |
| RH5ΔNL | Wright et al, 2014 | N/A |
| **Antibodies** | | |
| R5.034 | Barratt et al, 2023 | N/A |
| R5.016 | Alanine et al, 2019 | N/A |
| **Chemicals, enzymes and other reagents** | | |
| Ni-NTA agarose | Qiagen | 30210 |
| HiTrap™ Protein G HP column | Cytiva | GE29-0485-81 |
| Immobilised Papain | Thermo Fisher | 20341 |
| HiTrap™ rProtein A column | Cytiva | GE17-0403-03 |
| Superdex 200 Increase 10/300 column | Cytiva | GE28-9909-44 |
| Recombinant Protein A/G | Cytiva | 21186 |
| Matrix-M adjuvant | Novavax | N/A |
| **Software** | | |
| Rosetta Software Suite | Leman et al, 2020 | N/A |
| JalView | https://www.jalview.org | N/A |
| xia2-3dii | Gildea et al, 2022 | N/A |
| DIALS v3.0 | Beilsten-Edmands et al, 2020 | N/A |
| AIMLESS v0.73 | (Evans and Murshudov, 2013 | N/A |
| PHASER MR v2.8.3 | McCoy et al, 2007 | N/A |
| COOT v0.8.9.2 | Emsley and Cowtan, 2004 | N/A |
| BUSTER v2.10 | Bricogne G et al, 2017 | N/A |
| **Other** | | |
| CM5 series S sensor | Cytiva | BR100530 |

## Methods and protocols

### Design of the epitope mimics

The epitope mimics were designed using the Rosetta software suite (Leman et al, 2020). The region of PfRH5 containing the epitope was manually aligned with the scaffold (PDB: 3LHP) in Coot (Emsley and Cowtan, 2004), allowing a composite model to be generated in which residues 202–220 and 327–342 from PfRH5 were transplanted to the new epitope mimic. The resultant model was subjected to ab initio modelling using the energy minimisation function deployed in Rosetta. The folding process was carried out 10,000 times, scoring the output for stability. Next, the Rosetta package was used to perform computational site saturation mutagenesis for all residues in the synthetic immunogen other than those specified as invariant due to their participation in the epitope. This process generated 500 designs, which were ranked by their Rosetta score as well as their root-mean-square deviation from the desired design. Of the 500 designs, 447 were in a single cluster of closely related Rosetta scores and showed a Cα RMSD of <1 Å from the original design. The best 100 designs were then analysed by evolutionary trace analysis in JalView, allowing the selection of nine designs which best-represented sequence diversity for the designed proteins.

### Expression and purification of epitope mimics

The epitope mimics were each expressed with an N-terminal tag consisting of His$_6$-tag–thrombin cleavage site–Spy tag–TEV cleavage site. This allowed the expression of proteins which could be purified using metal ion affinity and could be cleaved to either remove all tags, or to leave a spy tag at the N-terminus to allow conjugation to virus-like particles which display the spy-catcher protein (Brune et al, 2016; Marini et al, 2019). The genes were inserted into the pEt15b vector (Novagen) and were transformed into Shuffle T7 express competent cells (New England Biolabs). Protein expression was induced when cells reached an optical density at 600 nm of 0.6 by addition of IPTG to a final concentration of 0.5 mM and cells were harvested after a further 4 h at 37 °C. Cells were lysed by sonication and proteins were purified using Ni-NTA affinity chromatography (Qiagen).

### Expression and purification of Fab and scFv fragments

The monoclonal antibody R5.034 (Barrett et al, 2024) was transiently expressed using Expi293F™ cells with the Expi293™ Expression System Kit (Thermo Fisher). Culture supernatants were harvested and passed through a 0.45-μm filter. The antibody was purified using a pre-packed 1 ml HiTrap™ Protein G HP column (Cytiva). Fab fragments were prepared by cleavage with immobilised Papain (20341, Thermo Scientific) overnight at 37 °C, and Fc and Fab fragments were separated using a pre-packed 1 ml HiTrap™ rProtein A column (Cytiva). The unbound fraction containing the Fab was retained and exchanged into PBS.

Single-chain variable fragments of R5.016 (Alanine et al, 2019) and R5.034 were constructed by linking the heavy and light variable chains using linkers of 18- and 15-residues, respectively (GGSSRSSSSGGGGSGGGG for R5.016 and GGGGSGGGGS GGGGS for R5.034). These were cloned into the pHLsec vector and were expressed transiently in FreeStyle™ 293-F cells (Thermo Fisher) in FreeStyle™ F17 Expression Medium supplemented with L-glutamine and 1× MEM non-essential amino acids (Gibco).

Cultures were harvested after 5 days (R5.016) or 6 days (R5.034), and supernatants were adjusted to pH 8.0 and 0.45 μm filtered before incubating with Ni-NTA resin equilibrated in 25 mM Tris pH 8.0, 150 mM NaCl (TBS). After washing the resin with 10-column volumes of TBS, followed by 20-column volumes of 25 mM Tris pH 8.0, 500 mM NaCl, 20 mM imidazole, bound proteins were eluted with TBS with 500 mM imidazole. To isolate monomeric scFv, eluted proteins were further purified by gel filtration on a Superdex 75 10/300 column (R5.016) or a Superdex 200 Increase 10/300 column (R5.034) into 20 mM HEPES pH 7.5, 150 mM NaCl.

### Expression of PfRH5ΔNL

RH5ΔNL (Wright et al, 2014) (a construct of RH5 encompassing residues K140-K247 and N297-Q526, thereby lacking its flexible N-terminus and internal loop) was expressed and secreted from a stable S2 cell line (ExpreS²ion Biotechnologies) in EX-CELL® 420 Serum Free Medium (Sigma-Aldrich) (Wright et al, 2014). After 3–4 days, the culture supernatant was harvested and adjusted to pH 8 with Tris, centrifuged at 9000× $g$ for 15 min and 0.45-μm filtered, then incubated with Ni Sepharose™ excel resin (Cytiva) for 2 h. Beads were washed with 5-column volumes of TBS, and 20-column volumes of wash buffer (25 mM Tris pH 8.0, 500 mM NaCl, 20 mM imidazole), then bound proteins eluted with elution buffer (TBS + 500 mM imidazole). Eluted proteins were diluted 1:1 in ConA binding buffer (20 mM Tris pH 7.5, 500 mM NaCl, 1 mM MnCl$_2$, 1 mM CaCl$_2$), then incubated with ConA Sepharose 4B resin (Cytiva) overnight at 4 °C, after which the unbound fraction containing RH5ΔNL was recovered. RH5ΔNL was further purified by gel filtration using an S200 Increase 10/300 column into SEC buffer (20 mM HEPES pH 7.5, 150 mM NaCl, 5% glycerol).

### Analytical size-exclusion chromatography

Analytical size-exclusion filtration was performed with a Superdex 75 Increase 10/300 column (Cytiva) in 20 mM HEPES pH 7.5 and 150 mM NaCl.

### Circular dichroism

Circular dichroism experiments were conducted using a Jasco J815 spectrophotometer. Proteins were buffer exchanged into 20 mM sodium phosphate pH 7.5, 20 mM NaF using PD-10 columns (Cytiva) and adjusted to a final concentration of 0.1 mg/ml. For each sample, 10 scans were performed at 20 °C from 260 nm to 190 nm wavelength, with measurements taken every 0.5 mm. A baseline determined using buffer alone was subtracted.

For thermal melt experiments, spectra were taken from 260 nm to 190 nm wavelength at 2 °C intervals from 20 °C to 90 °C with measurements taken every 0.2 nm. A baseline measurement for buffer alone at 20 °C was subtracted from all spectra.

### Surface plasmon resonance

Surface plasmon resonance experiments were carried out using a Biacore T200 instrument (Cytiva) with a buffer of 20 mM HEPES pH 7.5, 150 mM NaCl, 0.005% Tween-20. Recombinant Protein A/G (Pierce) was coupled to a CM5 chip by amine coupling. Antibody was captured onto the chip through binding to Protein A/G, and the epitope mimics were then flowed over at 30 μL/min with a contact time of 240 s and dissociation time of 400 s. After each cycle, the surface was regenerated using 10 mM using glycine pH 2.

Twofold dilution series from 500 nM to 3.9 nM were studied for each sample. Data were analysed using the BiaEvaluation software.

### Crystallisation and structure determination

To determine the structure of RH5-34EM bound to R5.016 scFv, components were combined at a 1:1 molar ratio, incubated for 1 h and complex was purified by size-exclusion chromatography using a Superdex 75 10/300 column in 20 mM HEPES pH 7.5, 150 mM NaCl. Crystals were obtained at 10 mg/ml in sitting drops by vapour diffusion using a well solution of 20% PEG8000, 0.1 M HEPES pH 7.0 after mixing 100 nl protein and 100 nl reservoir solution. They were cryoprotected through transfer into drops of well solution supplemented with 25% glycerol prior to cryocooling for data collection in liquid nitrogen. Data were collected at Diamond Light Source at the I04 beamline and were indexed using DIALS (v3.0) (Beilsten-Edmands et al, 2020) and scaled using AIMLESS (v0.73) (Evans and Murshudov, 2013) giving a dataset at a resolution of 1.63 Å. The structure was solved using molecular replacement using Phaser MR (McCoy et al, 2007) (v2.8.3), with the VH and VL domains of the scFv (PDB: 4U0R) (Wright et al, 2014) as search models. The model was built and refined using cycles of COOT (Emsley and Cowtan, 2004) (v0.8.9.2) and BUSTER (Bricogne G et al, 2017) (v2.10).

To determine the structure of RH5-34EM bound to R5.034 scFv, components were mixed at approximately equimolar ratio and complex was isolated by gel filtration using an Superdex 75 Increase 10/300 in 25 mM Tris pH 7.4, 150 mM NaCl. Crystals were obtained at 15 mg/mL in sitting drops by vapour diffusion at 18 °C in Morpheus A1 (0.1 M Buffer System 1 pH 6.5, 0.06 M Divalents, 30% v/v Precipitant Mix 1) after mixing 100 nl protein and 100 nl reservoir solution. Data were collected at Diamond Light Source on beamline I24 (wavelength 0.9999 Å). Diffraction data were processed using xia2-3dii(Gildea et al, 2022) to 1.75 Å. Molecular replacement was performed in PHASER (McCoy et al, 2007) using the RH5-34EM and scFv scaffold from the RH5-34EM:R5.016 structure, determined here, as search models. The structural model was further built and refined using COOT (Emsley and Cowtan, 2004) (0.9.3) and BUSTER (Bricogne G et al, 2017) (2.10.4). Two copies of the RH5-34EM:scFv complex are present in the unit cell.

To determine the structure of PfRH5 bound to R5.034, RH5ΔNL was treated with 1 µg/ml Endoproteinase Glu-C (Sigma) overnight at room temperature, then mixed with the R5.034 Fab fragment at slight molar excess. The complex was isolated by gel filtration using an Superdex 200 Increase 10/300 column into 20 mM HEPES pH 7.5, 150 mM NaCl, 5% glycerol. Crystals were obtained at 12.35 mg/mL in sitting drops by vapour diffusion at 18 °C in JCSG + A9 (0.2 M ammonium chloride, 20% w/v PEG 3350) containing Silver Bullets C10 (0.16% w/v β-Cyclodextrin, 0.16% w/v D-(+)-Cellobiose, 0.16% w/v D-(+)-Maltotriose, 0.16% w/v D-(+)-Melezitose hydrate, 0.16% w/v D-(+)-Raffinose pentahydrate, 0.16% w/v Stachyose hydrate, 0.02 M HEPES sodium pH 6.8) after mixing 100 nl protein, 100 nl reservoir solution and 50 nl additives. Data were acquired at the Synchrotron SOLEIL on beamline PROXIMA-1 (wavelength 0.9786 Å). Diffraction data were processed using DIALS (Beilsten-Edmands et al, 2020) and AIMLESS (Evans and Murshudov, 2013) in the CCP4 suite to 2.4 Å. Molecular replacement was performed using PHASER (McCoy et al, 2007) using a crystal structure of PfRH5 (PDB: 6RCU), the scFv of R5.034 (determined here) and

the constant domain of a Fab fragment (PDB: 4LLD) as search models. The model was further built using COOT (Emsley and Cowtan, 2004) (0.9.3) and refined using BUSTER (Bricogne G et al, 2017) (2.10.4) then PHENIX (Liebschner et al, 2019) (1.20.1). The unit cell contains one copy each of PfRH5 and R5.034 Fab fragments. The heavy-chain constant domain is less well resolved than the rest of the complex.

### Immunisation of rats

Rat immunisations were performed at Noble Life Sciences, Inc. (Maryland, USA). Female Wistar IGS rats ($n = 6$ per cohort) between 150 and 200 g (8–12 weeks old) were injected intramuscularly (IM) with antigens equimolar to 2 µg of soluble RH5.1 protein (full-length PfRH5 (Jin et al, 2018)). For RH5-34EM, this is 1.8 µg of VLP-conjugated immunogen, based on an observed coupling density of 100%. All immunogens were formulated in 25 µg of Matrix-M adjuvant (Novavax) with the cohort size selected based on previous similar studies. Blood samples were taken on days −2, 28, 56 for serum preparation and rats were terminally bled on day 70. Serum samples were then frozen and shipped to the University of Oxford, UK for testing. Each rat was analysed individually by ELISA and GIA. Blinding was not used.

### ELISA measurements

Nunc Maxisorp plates (Thermo Fisher Scientific) were coated overnight (4 °C, >16 h) with 50 µl/well of PfRH5.1 or RH5-34EM at 2 µg/mL diluted in Dulbecco's PBS (DPBS). The coated plates were washed in 6 times with wash buffer (PBS with 0.05% Tween-20; PBST) and blocked with 200 µL/well of RT Starting Block T20 (Thermo Fisher Scientific) for 1 h. Blocked plates were washed 6 times in DPBS/T and dilutions of the reference serum (serum induced by immunisation of rats with three doses of 2 µg PfRH5.1 in Matrix M in a previous study) and test serum prepared in Starting Block T20 were added with 50 µL/well and left for 2 h at RT. After incubation with serum, the plates were washed 6 times in DPBS/T and 50 µL of goat anti-rat secondary antibody (Sigma, A8438; 1:1000 prepared in Starting Block T20) was pipetted per well and left for 1 h at RT. Plates were washed six times with DPBS/T and 100 µL development buffer (p-nitrophenyl phosphate substrate diluted in diethanolamine buffer) was added per well and developed according to internal controls (average OD = 1.0 at 405 nm wavelength, approx. range 0.8–1.2). All serum samples were tested in triplicate against each coating antigen. The test serum samples were diluted to achieve OD 405 nm reading in the linear part of the standard curve. ELISA arbitrary units were converted to antigen-specific IgG concentration using a conversion factor determined by concentration-free calibration analysis, as described (Williams et al, 2024).

### IgG purification from rat serum

IgG was purified for growth-inhibitory activity (GIA) experiments, thereby removing non-specific effects due to serum components and enabling the concentration of IgG samples to allow saturation of growth-inhibition responses in positive controls. Serum samples (~3–4 ml) were diluted by adding binding buffer (IgG binding buffer, Thermo Scientific) to give a final volume of 8-10 ml. and were passed through a 2 ml protein G column (Immunopure Plus Immobilised Protein G Gel, Pierce) pre-equilibrated with 10 ml binding buffer. After applying the flow-through over the column

### The paper explained

#### Problem

Malaria still causes over 600,000 deaths and hundreds of millions of cases each year and improved vaccine immunogens are urgently required. The parasites that cause malaria replicate within human blood cells. If we can prevent red blood cell invasion, we can prevent the symptoms and transmission of the disease. The PfRH5 protein has been identified as a promising candidate for inclusion in vaccines as it is essential for blood cell invasion. However, blocking invasion requires high concentrations of growth-inhibitory antibodies. Can we design an improved PfRH5-based vaccine immunogen which induces a sufficiently growth-inhibitory response to prevent blood-stage malaria?

#### Results

The most effective growth-inhibitory monoclonal antibodies that target PfRH5 bind to a single epitope. Here, we solved the crystal structure of PfRH5 bound to the best-in-class human monoclonal antibody R5.034, which shares this epitope. We grafted the R5.034 epitope onto a three-helical scaffold and used rational protein design to successfully generate a synthetic immunogen, RH5-34EM, which binds R5.034 with high affinity. In a pre-clinical model, we show that RH5-34EM induces a high-quality, albeit low-quantity, immune response which prevents parasite growth at low PfRH5-specific antibody concentrations. Testing different vaccine regimens shows that a first dose of RH5-34EM, followed by two doses of PfRH5, induces the most effective growth-inhibitory antibody response.

#### Impact

We have designed a novel synthetic malaria vaccine immunogen which induces a high-quality parasite growth-inhibitory response. This is now available for clinical testing, either alone or in combination with vaccine components targeting other parts of the malaria life cycle.

NHS Blood and Transplant service. On day 1 of the assay, 20 ml of 3D7 *Plasmodium falciparum* culture in complete medium (incomplete medium [RPMI, 1% L-glutamine, 0.005% hypoxanthine, 25 mM HEPES], 10% heat-inactivated filtered pooled human serum from O+ individuals, 10 μg/ml gentamycin) at 2% haematocrit were sorbitol-synchronised. On day 2, 20 μl of synchronised parasites in 2x complete medium, 0.4% parasitaemia, 2% haematocrit were added to the twelve wells containing 20 μl 10 mM EDTA or incomplete medium or to the triplicate wells containing rat serum-purified IgG dilutions, 20 μl positive controls (40 μg/ml 2AC7, 30 μg/ml R5.016, 5 μg/ml R5.034), or 20 μl of a negative control monoclonal antibody (500 μg/ml EBL040) in incomplete medium. A parallel tracker *Plasmodium falciparum* culture (0.2% final parasitaemia) was also prepared. The assay plates and the tracker culture were then placed in a modular incubator and cultured at 37 °C for 2 days. On day 4, the assay plates were washed twice with cold PBS, and erythrocytes were resuspended by shaking.

The inhibition of *Plasmodium falciparum* growth was measured using the lactate dehydrogenase (LDH) assay, with 120 μl LDH substrate containing 3-acetylpyridine adenine dinucleotide (50 μg/ml), diaphorase (1 U/ml), and nitro blue tetrazolium (0.2 mg/ml) added to each well. The assay plates were developed until reaching an optical density of 0.4–0.6 in the wells containing parasitized erythrocytes and read at 650 nm. Percent inhibition of *Plasmodium falciparum* growth was calculated using the following formula:

$$\%inhibition = \left( \frac{(A650\ immune\ sample - A650\ erythrocytes\ only)}{(A650\ pre-immune\ control - A650\ erythrocytes\ only)} \right) \times 100\%$$

three more times, it was washed twice with 10 ml binding buffer. Bound IgG was eluted with 10 ml of elution buffer (0.1 M glycine pH 2.7) into 350 μl 1 M Tris pH9. The eluate was buffer exchanged once into RPMI1640 (Sigma) using 15 ml 30 K molecular weight cut-off Amicon Ultra Centrifugal filter device (Fisher Scientific UK Ltd) and then into incomplete medium (RPMI, 1% L-glutamine, 0.005% hypoxanthine, 25 mM HEPES) with final volume around 250 μl. For GIA, purified IgG were clarified by adding 50 μl of 100% haematocrit human O+ red blood cells (RBCs) for each 1 ml of the original serum sample that was used to prepare the IgG. The sample was mixed by gentle inversion for an hour at room temperature. Non-adsorbed IgG was collected by centrifuging in a sterile spin filter (Costar Spin-X Centrifuge Tube Filters 0.22 μm Pore, Scientific Laboratory Supplies Limited) at 5000×*g* for 2 min to pellet RBC and retaining the supernatant.

The concentration of total IgG was determined using absorbance of 280 nm wavelength light. To convert total IgG into PfRH5-specific IgG, we first performed an ELISA assay using PfRH5-coated wells. To convert ELISA absorbance into concentration, we used a conversion factor that had been determined by concentration-free calibration analysis, as described in reference (Williams et al, 2024).

### Growth-inhibitory activity measurement

Erythrocytes from O+ individuals for 3D7 *Plasmodium falciparum* culture and growth-inhibition assays (GIA) were obtained via the

## Data availability

Crystal structures are available at the Protein Data Base with accession codes 8RZ0, 8RZ1 and 8RZ2 and the remaining data is provided as source data.

The source data of this paper are collected in the following database record: biostudies:S-SCDT-10_1038-S44321-024-00123-0.

## Peer review information

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

## Acknowledgements

This work was funded by Wellcome Investigator Awards (101020/Z/13/Z and 220797/Z/20/Z) to MKH and, in part by the United States Agency for International Development (USAID) Malaria Vaccine Development Program (7200AA20C00017). The findings and conclusions are those of the authors and do not necessarily represent the official position of USAID. The authors thank colleagues at PATH (Ashley Birkett, Randall MacGill and Allison Clifford) and USAID (Lorraine Soisson and Robin Miller) for comments on the manuscript. TEH was funded by the Peter J. Braam Scholarship for Global Wellbeing at Merton College, Oxford. The authors thank Sumi Biswas for the provision of the HBsAg virus-like particle. The authors thank Ed Lowe and the beamline scientists at Diamond Light Source and Soleil for support with crystallographic data collection and David Staunton for help with biophysical analysis. NHS Blood & Transplant (NHSBT) have provided material in support of this research but the views expressed are not necessarily those of NHSBT.

## Author contributions

**Thomas E Harrison**: Conceptualisation; Formal analysis; Validation; Investigation; Visualisation; Methodology; Writing—review and editing. **Nawsad Alam**: Conceptualisation; Formal analysis; Investigation; Methodology; Writing—review and editing. **Brendan Farrell**: Formal analysis; Validation; Investigation; Methodology; Writing—review and editing. **Doris Quinkert**: Investigation. **Amelia M Lias**: Investigation. **Lloyd D W King**: Investigation. **Lea K Barfod**: Investigation. **Simon J Draper**: Investigation. **Ivan Campeotto**: Conceptualisation; Investigation; Methodology; Writing—review and editing. **Matthew K Higgins**: Conceptualisation; Formal analysis; Funding acquisition; Investigation; Methodology; Writing—original draft; Project administration.

Source data underlying figure panels in this paper may have individual authorship assigned. Where available, figure panel/source data authorship is listed in the following database record: biostudies:S-SCDT-10_1038-S44321-024-00123-0.

## Disclosure and competing interests statement

MKH, TEH, NA, DQ, AML, LDWK, SJD and IC are inventors of patents related to vaccines and/or monoclonal antibodies which target PfRH5. SJD is a co-founder of and shareholder in SpyBiotech.

# Expanded View Figures

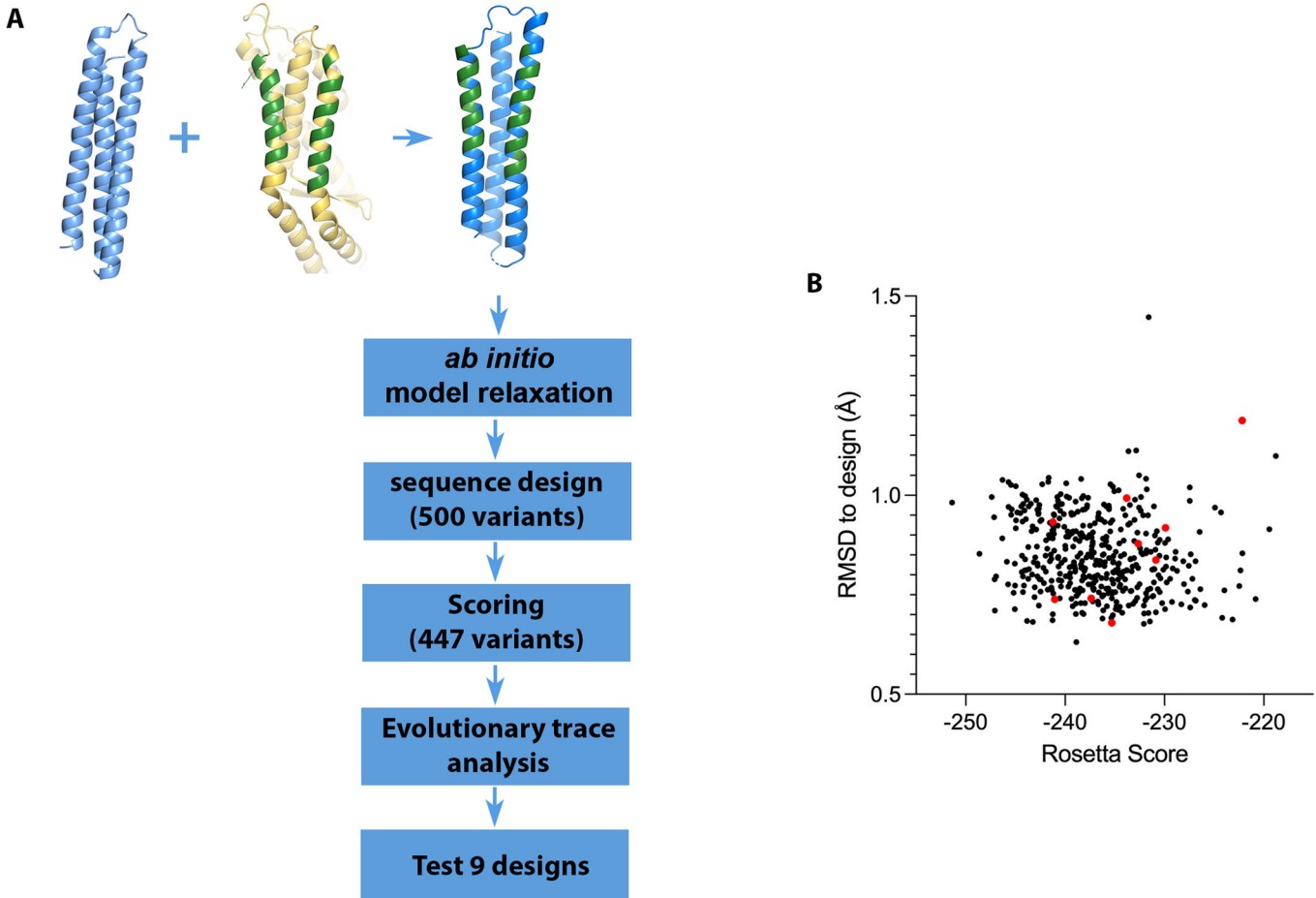

**Figure EV1. Design process.**

(A) A schematic showing the design of RH5-34EM. The 9AD4 epitope (green) of PfRH5 (yellow) was grafted onto a synthetic three-helical bundle scaffold (blue), generating an initial design, followed by a Rosetta-based design strategy. (B) A plot of Rosetta score against predicted root-mean square deviation to the design of the 447 sequence variants passing the scoring criteria. The nine red circles are for the designs taken forward for testing (designs 1–9).

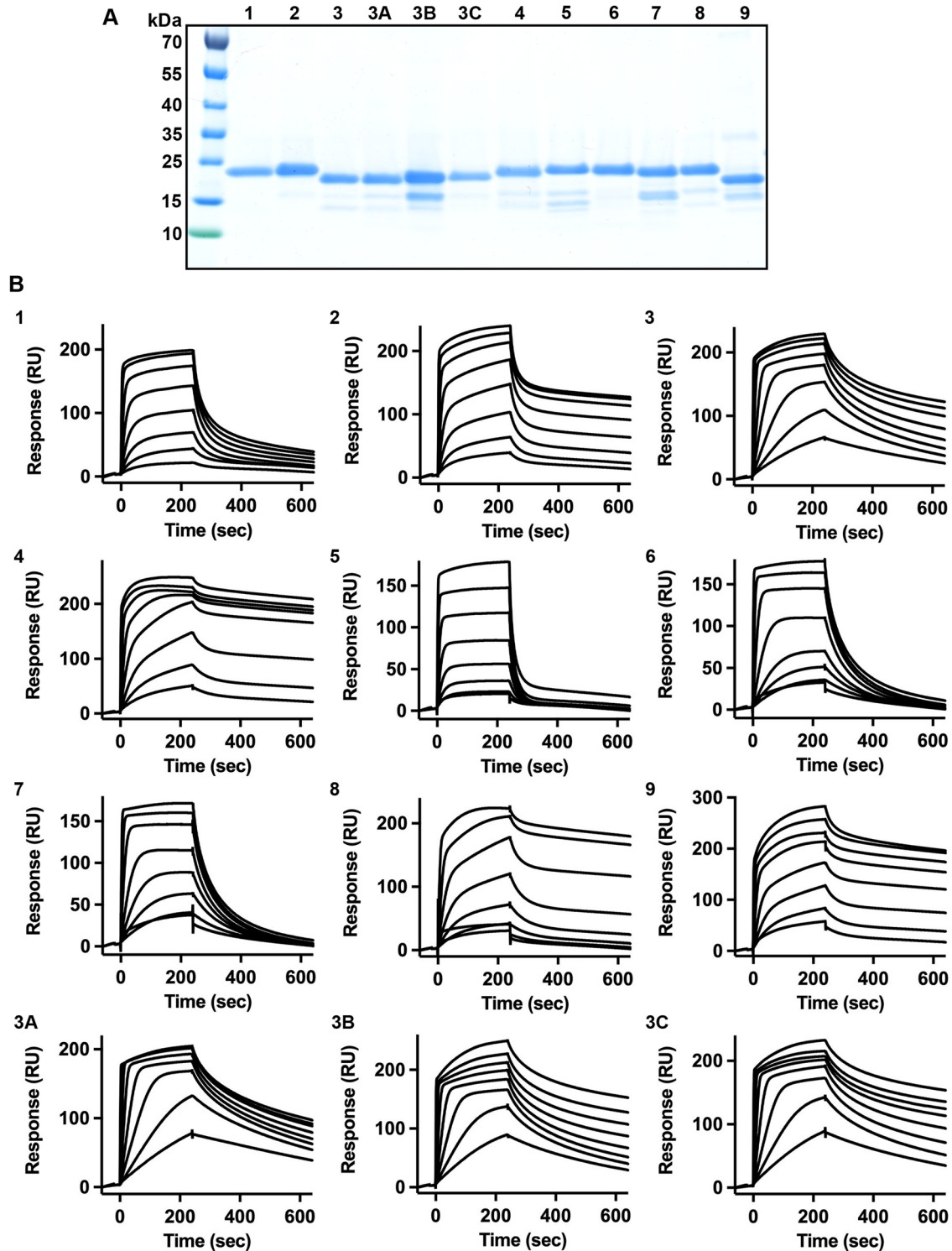

**Figure EV2.  Characterisation of epitope mimic designs.**

(A) SDS PAGE gel for the twelve designs, stained with Coomassie. The designs vary in molecular weight from 16.2 to 16.5 kDa. (B) Surface Plasmon Resonance traces for the twelve designs. In each case, antibody 9AD4 was captured on the chip surface and a dilution series of each epitope mimic, from a maximum concentration of 500 nM, was flowed over this surface.

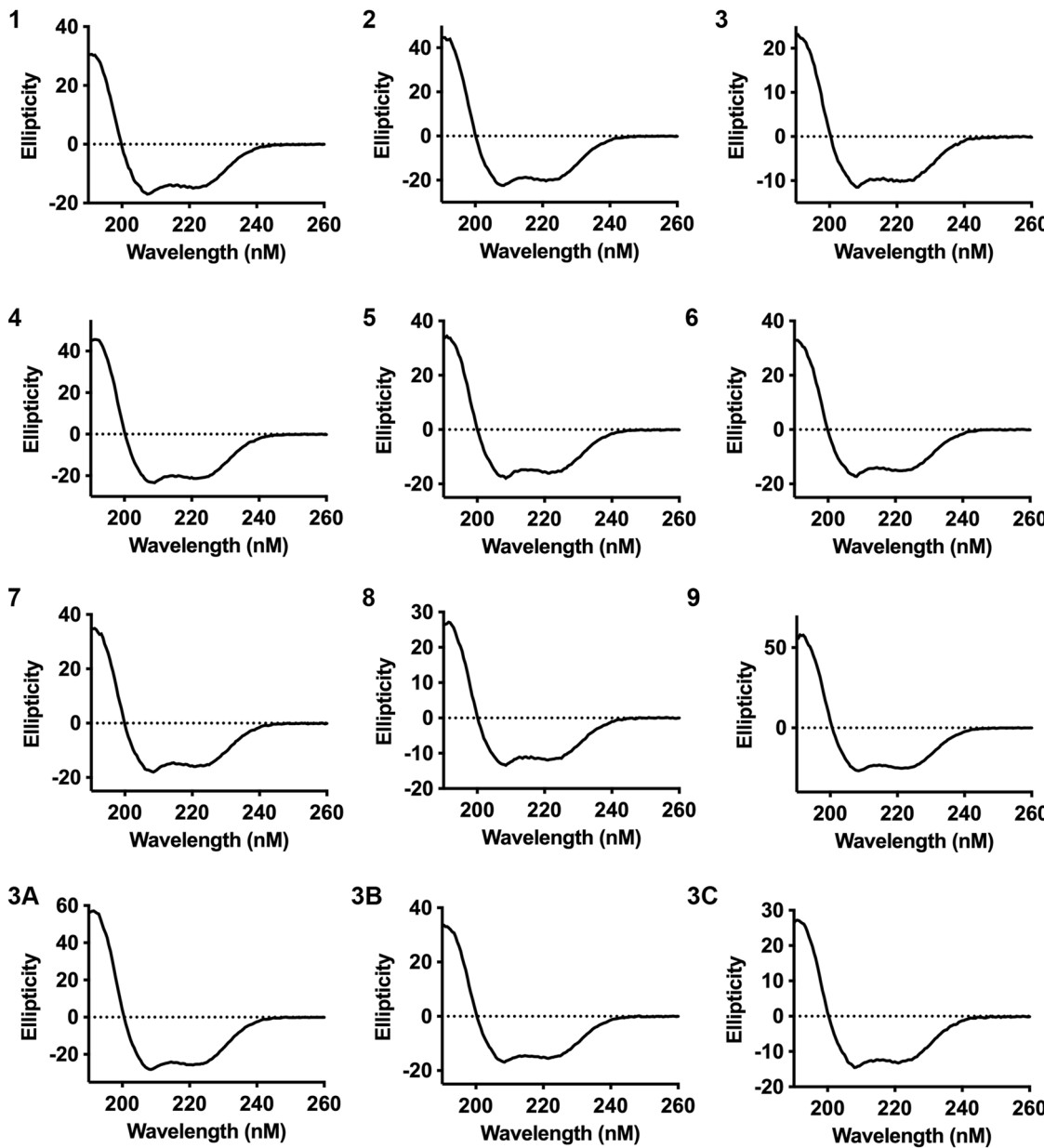

**Figure EV3.  Circular dichroism traces for the twelve designs.**

Circular dichroism measurements for the twelve different epitope mimic designs.

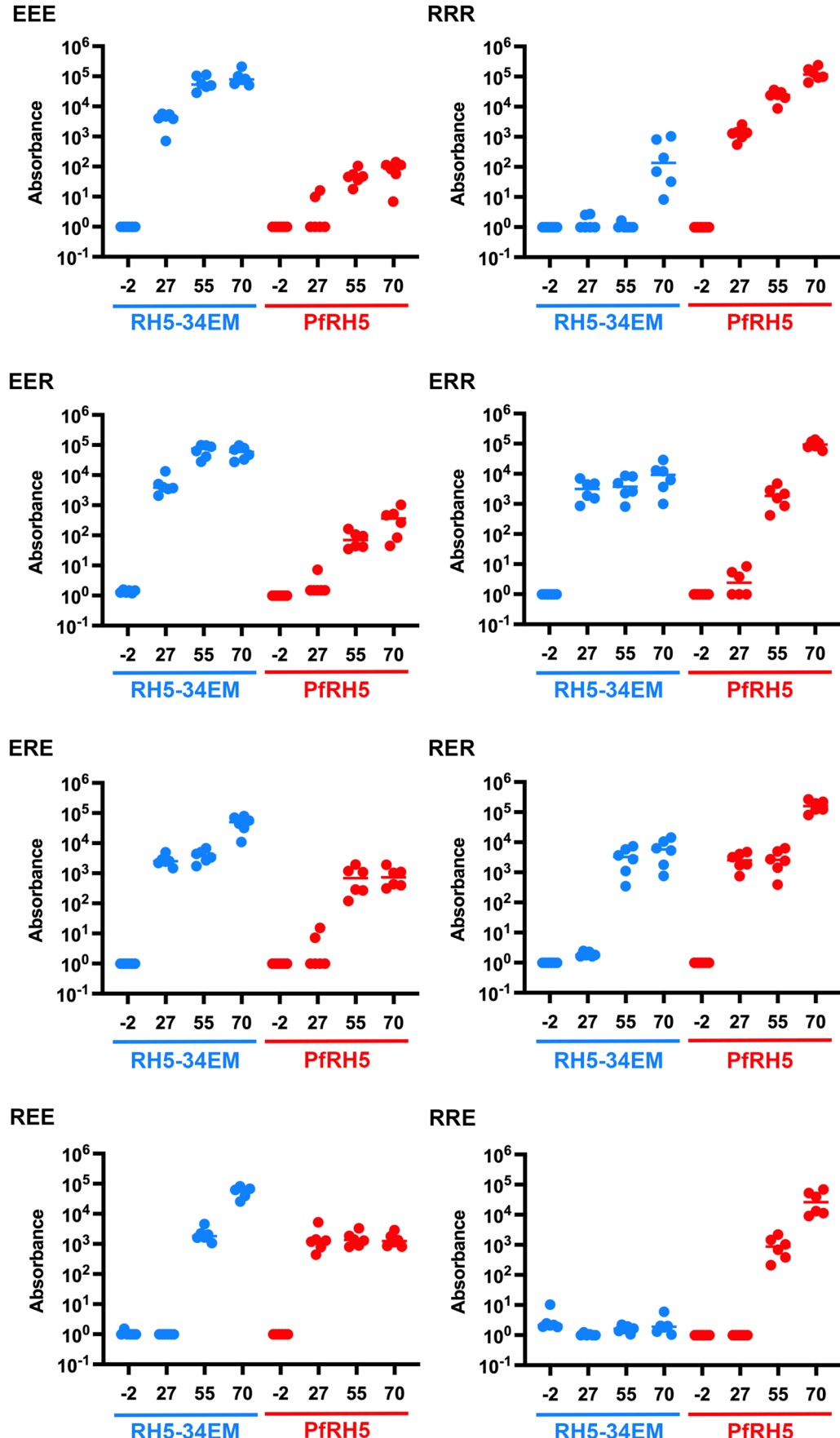

◄ **Figure EV4. ELISA data for different time points during immunisation.**

Absorbance measurements of sera against RH5-34EM (blue) and PfRH5 (red) at day -2 (before immunisation) and days 27, 55 and 70 (after the first, second and third vaccine doses). The plots show the eight different vaccine regimens. In each cohort we immunised 6 rats and analysed individually and measured absorbance once for each sample.

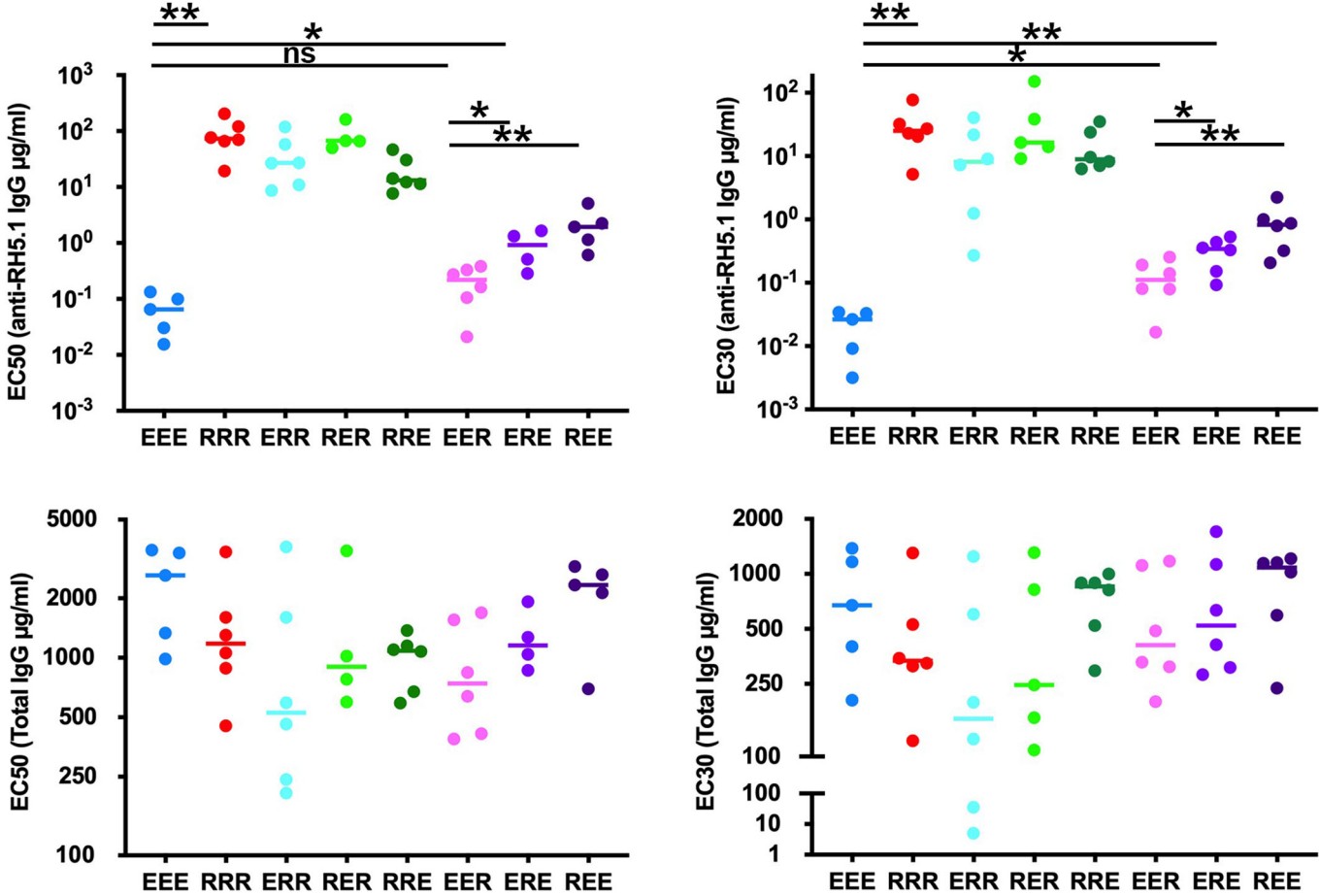

**Figure EV5. Comparing EC$_{50}$ and EC$_{30}$.**

Growth-inhibitory activity measurements, as shown in Fig. 5, were analysed to extract EC$_{30}$ and EC$_{50}$ values. The EC$_{30}$ plots here replicate those shown in Fig. 5. In addition, we show plots of EC$_{50}$ for all data points for which growth-inhibitory activity reaches this level. In both cases, we provide data for total IgG and for PfRH5-specific IgG. Statistical significance determined using a two-tailed Mann–Whitney test corrected for multiple comparisons (* indicates <0.05 and ** indicates <0.005). For EC30, for EEE vs RRR, $P = 0.0043$; for EEE vs EER, $P = 0.0303$; for EEE vs ERE, $P = 0.0079$; for EER vs ERE, $P = 0.0411$; for EER vs REE, $P = 0.0043$. For EC50, for EEE vs RRR, $P = 0.0043$; for EEE vs EER, p = 0.0823; for EEE vs ERE, $P = 0.0159$; for EER vs ERE, $P = 0.0381$; for EER vs REE, $P = 0.0043$.

