## [Peer Review File · EMBO Molecular Medicine]

Rational structure-guided design of a single epitope blood stage malaria vaccine immunogen

Thomas Harrison, Nawsad Alam, Brendan Farrell, Doris Quinkert, Amelia Lias, Lloyd King, Lea Barfod, Simon Draper, Ivan Campeotto, and Matthew Higgins

Corresponding authors: Matthew Higgins (matthew.higgins@bioch.ox.ac.uk) , Matthew Higgins (matthew.higgins@bioch.ox.ac.uk), Ivan Campeotto (Ivan.Campeotto@nottingham.ac.uk)

Review Timeline:

Submission Date:	25th Mar 24
Editorial Decision:	22nd Apr 24
Revision Received:	10th Jul 24
Editorial Decision:	13th Aug 24
Revision Received:	20th Aug 24
Accepted:	20th Aug 24

Editor: Zeljko Durdevic

Transaction Report:

22nd Apr 2024

Dear Dr. Higgins,

Thank you for the submission of your manuscript to EMBO Molecular Medicine. We have now received feedback from the three reviewers who agreed to evaluate your manuscript. All three referees support publication of the manuscript but also raise important criticism that should be addressed in a revision of the current manuscript. If you would like to discuss further the points raised by the referees, I am available to do so via email or video. Let me know if you are interested in this option.

Further consideration of a revision that addresses reviewers' concerns in full will entail a second round of review. EMBO Molecular Medicine encourages a single round of revision only and therefore, acceptance or rejection of the manuscript will depend on the completeness of your responses included in the next, final version of the manuscript. For this reason, and to save you from any frustrations in the end, I would strongly advise against returning an incomplete revision. Also, please format the manuscript according to our "Author Guidelines": <https://www.embopress.org/page/journal/17574684/authorguide>

We would welcome the submission of a revised version within three months for further consideration. Please let us know if you require longer to complete the revision.

I look forward to receiving your revised manuscript.

Yours sincerely,

Zeljko Durdevic

We require:

- 1) A .docx formatted version of the manuscript text (including legends for main figures, EV figures and tables). Please make sure that the changes are highlighted to be clearly visible.
- 2) Individual production quality figure files as .eps, .tif, .jpg (one file per figure). For guidance, download the 'Figure Guide PDF': (<https://www.embopress.org/page/journal/17574684/authorguide#figureformat>).
- 3) A .docx formatted letter INCLUDING the reviewers' reports and your detailed point-by-point responses to their comments. As part of the EMBO Press transparent editorial process, the point-by-point response is part of the Review Process File (RPF), which will be published alongside your paper.
- 4) A complete author checklist, which you can download from our author guidelines (<https://www.embopress.org/page/journal/17574684/authorguide#submissionofrevisions>). Please insert information in the checklist that is also reflected in the manuscript. The completed author checklist will also be part of the RPF.
- 5) Please note that all corresponding authors are required to supply an ORCID ID for their name upon submission of a revised manuscript.
- 6) It is mandatory to include a 'Data Availability' section after the Materials and Methods. Before submitting your revision, primary

datasets produced in this study need to be deposited in an appropriate public database, and the accession numbers and database listed under 'Data Availability'. Please remember to provide a reviewer password if the datasets are not yet public (see <https://www.embopress.org/page/journal/17574684/authorguide#dataavailability>).

13) Author contributions: You will be asked to provide CRediT (Contributor Role Taxonomy) terms in the submission system. These replace a narrative author contribution section in the manuscript.

14) A Conflict of Interest statement should be provided in the main text.

15) Every published paper now includes a 'Synopsis' to further enhance discoverability. Synopses are displayed on the journal webpage and are freely accessible to all readers. They include a short stand first (maximum of 300 characters, including space) as well as 2-5 one-sentence bullet points that summarize the paper. Please write the bullet points to summarize the key NEW findings. They should be designed to be complementary to the abstract - i.e. not repeat the same text. We encourage inclusion of key acronyms and quantitative information (maximum of 30 words / bullet point). Please use the passive voice. Please attach these in a separate file or send them by email, we will incorporate them accordingly.

Please also suggest a striking image or visual abstract to illustrate your article as a PNG file 550 px wide x 300-800 px high.

**** Reviewer's comments ****

Referee #1 (Remarks for Author):

Rh5 as part of the merozoite invasion complex is an exceptional vaccine target due to its sequence conservation compared to other blood stage antigens. Nevertheless, the first vaccine design failed to induce highly protective antibody response in humans. The authors address the need to optimize the Rh5 immunogen design for better quality antibody responses by grafting the epitope of one of the most potent mAbs at the time on a scaffold. The authors confirm that the epitope overlaps with that of even more potent human mAbs that became available later.

Through a rational design and down-selection process, the authors developed a small synthetic immunogen with improved stability over previous designs that induced about 1000x more potent anti Rh5 antibody responses in rats when presented on nanoparticles than Rh5 in the same potent adjuvant. This change in quality of the response is impressive. However, the response was about 1000x lower and therefore the quality differences did not make up for the difference in quantity. Through heterologous immunization regimens, the authors determined that immunogenicity can be boosted with Rh5.

This is the first attempt to use structure-guided protein design strategies to develop an improved Rh5-based malaria vaccine. The data demonstrate that this strategy holds great promise. Although the authors have not developed second generation designs based on their findings, they provide a wealth of data including co-crystal structures, and the discussion addresses how the authors envision further improvements.

One aspect that I find confusing is the use of purified IgG rather than the use of diluted sera in the ELISA and GIA. Could the authors provide data from serially diluted sera for comparison, or why did they choose to purify the IgG? (How) were the IgG concentrations adjusted in the ELISA (and GIA)? A brief description in the methods section would be helpful.

It is also unclear why the total IgG data does not reflect the Rh5 calibrated response. The data suggest that the synthetic immunogen does not include all neutralizing epitopes on Rh5. This point is mentioned in the discussion, but the data is difficult to interpret. Could there be other serum factors (non-Rh5 antibodies) that mediate parasite inhibition or synergistic effects (as discussed)? If there is no clear answer, the total IgG data should be moved to the supplement for clarity. The authors state specifically that the aim was to improve the quality of the response using the 34-EM design, which should be the focus of the manuscript.

It is unconventional to show EC30 rather than EC50. Why is did the authors decide to show EC30 values?

The dilution series of sera from individual animals/groups do not show the complete sigmoid curve that is expected and necessary to accurately determine EC values. What is underlying?

Is there a way to indicate the relative proportion of the Rh5-34EM response induced by the different immunization regimens? It would help the reader to understand which response is mediating protection after homologous and heterologous E/R immunization.

In the heterologous prime-boost experiments, changes in titers might represent epitope shifts, especially after immunization with Rh5. Can this be assessed, e.g. by EMPM? It will be important to understand whether other neutralizing epitopes are targeted. The fact that the anti-Rh5-34EM response peaks after two immunizations and cannot be boosted could be due to the fact that epitope shifts cannot be induced given the size of the immunogen. This might also explain the inability to boost with a third immunization (although it would be interesting to see whether a delayed boost might overcome this).

Are the immunization data representative of independent experiments? An n=1 does not seem to be appropriate.

Specific comments:

ELISA:

How does the molarity of Rh5 vs Rh5-34EM compare and why was the OD limited to about 1? Could this affect the absorbance-based comparisons.

Can the authors provide examples of the raw data (similar to the GIA data) as part of the supplement?

Were all ELISA experiments that are shown in individual graphs performed in parallel? This seems to be essential given that absorbance data are shown. How many times were the ELISAs repeated?

Can the authors provide a brief description of how they purified the serum IgG (including the adjustment of concentrations or were these not adjusted?) and how they calibrated the Rh5 concentrations?

Suppl. Fig. 3:

The animals in the RRE group did not respond to the R prime (d27) compared to animals in groups RRR, REE and RER. Why is that? Is it because of the absorbance readout? The inconsistency needs to be addressed.

Referee #2 (Remarks for Author):

Summary

The manuscript by Harrison, Alam, Farrell et al. extends the molecular optimization of the PfRH5 antigen, which is the leading blood-stage vaccine candidate for malaria. This work presents a reverse-vaccinology approach, informed by previously reported structure-activity relationships of potent mAbs, as well as for R5.034 which is described in this paper, to immuno-focus the most potent RH5 epitope on a carrier scaffold. The authors use a combination of computational and biophysical approaches to downselect towards a lead antigen candidate, which they evaluate in rodent immunizations with comparisons to full-length RH5. The immunization data presented indicates that the novel engineered RH5 antigen can be multimerized to produce a focused, potentially inhibitory humoral response towards the target epitope. Importantly, different prime-boost regimens are assessed which suggest that while total sera potency may not be improved by this novel antigen, the quality of the antibodies generated is indeed higher than those induced by the full-length protein. The manuscript is an important contribution to the field of malaria blood-stage vaccine design and would be notably enhanced for publication by addressing the following points.

Major points

Rodent immunization studies were performed by comparing monomeric full-length RH5 (referred to as RH5.1) with RH5-34EM multimerized on the HBsAg VLP platform. The manuscript requires a more detailed explanation of the immunization doses used in this study, as it is unclear what "equimolar to 2 μ l of soluble RH5.1 protein" means. Additionally, the authors should more clearly indicate that they are comparing monomeric RH5.1 with multimerized RH5-34EM. Would PfRH5.1-Spy-tag-Spy-catcher-HBsAg not have been a better comparator arm, or perhaps a RH5-34EM monomeric arm in addition to its HBsAg-multimerized state as an important comparator to gauge the impact of multimerization on immunogenicity for this novel immunogen? If this data is not possible, the implications of comparing monomers vs multimers should be stated in the Discussion and these limitations of the study clearly discussed.

It is curious that three doses of RH5.1 will induce intermediate antibody titers against RH5-34EM protein as measured by ELISA (Figure 4c), while two doses of RH5.1 followed by one dose of RH5-34EM produces effectively no RH5-34EM-reactive sera (Figure 5a). It would be helpful for the readers' interpretation of the in vivo immunogenicity data for the authors to expand on whether they believe there is a molecular basis for this finding or if this is due to experimental variability.

The immunization data indicates that while the total IgG potency may not have been improved by RH5-34EM, the quality of the RH5-specific antibodies was improved compared to the full-length protein. The conclusion of this manuscript would meaningfully benefit from a discussion section focused on potential strategies to increase the sera titers for RH5-34EM.

Lines 203-205: what about EC50? How does the function of the purified IgG compare if this more stringent cut-off is used? This also applies to data reported in Fig. 5. It would be important to present EC50 data perhaps as a Supp Fig. and mention it in the Results, as it also provides a link to the best mAb potency stated in the Introduction.

Minor points

The authors should provide current malaria deaths from the World Malaria Report 2023.

Could the authors elaborate on the known genetic polymorphisms within the focused epitope selected for RH5-34EM?

It would be informative to the field of research related to antigen scaffolding for the authors to present ELISA data on the scaffold-specific responses induced by RH5-34EM, if available. This would also greatly enhance the understanding of the differences presented for the total IgG titers compared to the RH5-specific IgG titers.

To better understand the protein engineering workflow and in silico down-selection process, it would be helpful for the authors to provide a supplementary figure detailing the Rosetta pipeline and the predicted energies of the designs.

The expected protein molecular weights for constructs 1-9 and 3A-C should be indicated next to the SDS-PAGE in Supplementary figure 1a so that the reader can better interpret the migration distance and banding patterns presented on the gel.

Fig 1f: there are large error bars on KD measurements for constructs 3A, 3B and 3C (~50x range in error bars). Could this be improved? If not, can the authors comment on this high variability for this typically precise biophysical measurement?

Lines 173-174: the RH5-34EM nomenclature rationale was introduced above (lines 147-148), is it necessary to restate here again?

Lines 195-196: "It is therefore noteworthy that these two groups showed a similar ELISA reactivity against PfRH5 after three doses, [...]" This statement is misleading. Shouldn't it say: "similar ELISA reactivity against the opposite antigenic probes after three doses"? See Figure 4d that clearly shows the two different antigenic probes used.

The Rfree is quite high (30.46%) for the PfRH5:R5.034 complex despite decent data collection statistics at 2.4 Å resolution. Can the model be further improved, or can the authors provide an explanation for this relatively poor agreement between the model and unbiased X-ray data?

The authors should provide representative maps depicting electron density of antibody-antigen interfaces as a supplemental figure.

The authors should confirm that figure and table callouts are correct as several appear to reference incorrectly. Additionally, the panel orders should match the order of the call outs in the text.

There are several typos for punctuation and improper spacing between numbers and units throughout the text and figures.

Referee #3 (Comments on Novelty/Model System for Author):

The experiments were performed in the most logical manner possible.

Referee #3 (Remarks for Author):

Harrison, Alam, Farrell, and colleagues report the design and characterization of an epitope-scaffold immunogen based on the 9AD4/R5.016/R5.034 epitope in PfRH5. The manuscript is clearly written, the results are clearly presented in the figures, and the analysis and interpretation of the data is objective and balanced. I commend the authors on the last point. The manuscript is suitable for publication with only very minor revisions.

Major points:

-- Lines 9-11: I hesitate to call this a "major point", but in the absence of major flaws in the manuscript, there is one sentence in the Abstract that should be revised: "In immunised rats, the immunogen induced PfRH5-targeting antibodies that inhibit parasite growth at a thousand-fold lower concentration than those induced through immunisation with PfRH5." That's not quite true as written. The antibodies elicited by RH5-34EM were more potent in the GIA ****per unit PfRH5 binding antibody****. But this is neither surprising nor relevant in real-world vaccination scenarios. The sentence as written implies that the designed immunogen was 1000x more potent, and that simply isn't true. The authors should revise this sentence.

Minor points:

-- Lines 53-54: "epitope for a subdominant neutralising monoclonal antibody" should be "subdominant epitope for a neutralising monoclonal antibody".

-- Lines 104-107: "We selected design 3, which showed a symmetrical size exclusion chromatography profile, which bound 9AD4 with a high affinity and slow off rate, and which gave the closest predicted root-mean-square-deviation to the starting epitope configuration during the design process" should be "We selected design 3, which showed a symmetrical size exclusion

chromatography profile, bound 9AD4 with a high affinity and slow off rate, and gave the closest predicted root-mean-square-deviation to the starting epitope configuration during the design process."

-- Lines 138-142: The conclusion as stated is not supported by the data. "indicate that our epitope mimics... will... induce R5.016 and R5.034" should be "suggest that..." or "indicate that... may induce R5.016- and R5.034-like antibodies."

-- Fig. 2d-f: What is the difference between the blue and red dashed lines? This should be defined in the figure legend.

-- Lines 179-181: But PfRH5.1 was not conjugated to HBsAg? Is that a fair comparison? The authors should clarify this point and justify comparing a particulate to a non-particulate immunogen.

-- Fig. 4b-d: Is this really Absorbance on the y axis, or is it a reciprocal titer (i.e., ED50)?

-- Lines 195-198: You cannot quantitatively compare ELISA titers against two different antigens laid down on the ELISA plate. Binding to the plate, and therefore the apparent titer, can vary widely for different antigens. As a result, the authors' claim here is invalid. If the authors wish to support this claim, they should determine the absolute concentration of antibody specific to each antigen by using a standard curve in their ELISAs. Otherwise, the authors should remove this claim.

-- Line 270: This is "Reverse vaccinology 2.0" (Rappuoli), or "Structure-based antigen design". The authors should consider citing Rappuoli or Burton's 2002 classic, "Antibodies, viruses, and vaccines", at the end of this sentence.

-- Line 298: "effective" should be "effectively".

Referee #1 (Remarks for Author):

Rh5 as part of the merozoite invasion complex is an exceptional vaccine target due to its sequence conservation compared to other blood stage antigens. Nevertheless, the first vaccine design failed to induce highly protective antibody response in humans. The authors address the need to optimize the Rh5 immunogen design for better quality antibody responses by grafting the epitope of one of the most potent mAbs at the time on a scaffold. The authors confirm that the epitope overlaps with that of even more potent human mAbs that became available later.

Through a rational design and down-selection process, the authors developed a small synthetic immunogen with improved stability over previous designs that induced about 1000x more potent anti Rh5 antibody responses in rats when presented on nanoparticles than Rh5 in the same potent adjuvant. This change in quality of the response is impressive. However, the response was about 1000x lower and therefore the quality differences did not make up for the difference in quantity. Through heterologous immunization regimens, the authors determined that immunogenicity can be boosted with Rh5.

This is the first attempt to use structure-guided protein design strategies to develop an improved Rh5-based malaria vaccine. The data demonstrate that this strategy holds great promise. Although the authors have not developed second generation designs based on their findings, they provide a wealth of data including co-crystal structures, and the discussion addresses how the authors envision further improvements.

We thank the reviewer for their fair and positive assessment.

One aspect that I find confusing is the use of purified IgG rather than the use of diluted sera in the ELISA and GIA. Could the authors provide data from serially diluted sera for comparison, or why did they choose to purify the IgG? (How) were the IgG concentrations adjusted in the ELISA (and GIA)? A brief description in the methods section would be helpful.

As the reviewer suggests, we conducted ELISA measurements using dilutions of sera and there were some errors in the figure legends in the original manuscript which made this confusing. We have now corrected these errors in the legends for Figures 4 and 5 and Figure EV3.

The GIA assays were performed with IgG as is standard for this assay. See for example Silk et al (2024) Lancet Infectious Disease for human studies and Williams et al (2024) Nature Comms for rat studies. This avoids the introduction of non-specific effects due to serum components. In addition, the constraints of the assay require antibody concentrations which would not be achievable using sera but are possible through concentration of purified IgG. We have added a sentence to the methods (lines 531-533) to clarify this point.

We have now added methods sections on how we purified IgG and how we converted total IgG concentration into anti-PfRH5 IgG concentration. These are now found in lines 530-552. We apologise for the omission of these from the original manuscript.

It is also unclear why the total IgG data does not reflect the Rh5 calibrated response. The data suggest that the synthetic immunogen does not include all neutralizing epitopes on Rh5. This point is mentioned in the discussion, but the data is difficult to interpret. Could

there be other serum factors (non-Rh5 antibodies) that mediate parasite inhibition or synergistic effects (as discussed)? If there is no clear answer, the total IgG data should be moved to the supplement for clarity. The authors state specifically that the aim was to improve the quality of the response using the 34-EM design, which should be the focus of the manuscript.

Our view is that the comparison between the GIA of total IgG and PfRH5-specific IgG is extremely important. The total IgG, which contains non-PfRH5-binding antibodies, as well as those that bind to PfRH5, is most representative of sera and so the GIA of total IgG is the best representation of the likely protective effect of the vaccine in a vaccinated individual. For this reason, our view is that it is very important that we present the GIA from total IgG. We have clarified this point in lines 202-204. The knowledge that total IgG contains many antibodies which do not bind to PfRH5 is hopefully clearer now that we have provided a method for total IgG purification (lines 530-552). To understand the quality of the PfRH5-specific antibody response, it is then very important that we also present the effect of PfRH5-specific antibodies on GIA. We have clarified this point in lines 211-213. We are of the view that it is important to present all this data and hope that our changes to the manuscript have made the rationale for our experiments clearer for readers.

It is unconventional to show EC30 rather than EC50. Why did the authors decide to show EC30 values? The dilution series of sera from individual animals/groups do not show the complete sigmoid curve that is expected and necessary to accurately determine EC values. What is underlying?

Is there a way to indicate the relative proportion of the Rh5-34EM response induced by the different immunization regimens? It would help the reader to understand which response is mediating protection after homologous and heterologous E/R immunization.

Our decision to use EC30 rather than EC50 is a pragmatic one, due to the quantity of PfRH5-specific IgG obtained through rat immunisation. As immunisation with RH5-34EM induced a lower quantity of PfRH5-specific IgG, there was insufficient sample in all cases to reach EC50 in the GIA data. Rather than exclude samples, or pool samples, both of which would have led to a loss of information, we therefore selected to report EC30 for individual rats. We now also include the EC50 data in Figure EV5, for those rats which reached provided sufficient IgG to reach this level of inhibition. This gives a very similar outcome, with similar ranking of efficacy and similar statistics, giving confidence to our EC30 data.

Our view is that the failure to generate a sigmoidal curve is also due to sample quantity as we could not reach the higher concentrations required to complete the curves. We note that for samples which were not concentration limited (i.e. RRR and RRE in Figure 5), the curves are sigmoidal. Again, or desire not to pool samples, thereby losing information from individual rats, lies behind this.

To indicate the proportion of the RH5-34EM response in each of the prime-boost regimens, the best approach would be to deplete RH5-34EM specific antibodies from each serum sample and to re-run the GIA assays. We have done this in other more focused studies, in which we immunised rabbits with a smaller number of immunogens to obtain larger volumes of sera (i.e. [BioRxiv doi.org/10.1101/2024.06.23.600241](https://doi.org/10.1101/2024.06.23.600241)). but cannot envisage a way to achieve this with our current samples in this study. While we agree that this is an

interesting potential future study, we are not able to add this additional data here, as this would require an additional rat study. We are of the view that this data is not required for us to draw conclusions that we make here.

In the heterologous prime-boost experiments, changes in titers might represent epitope shifts, especially after immunization with Rh5. Can this be assessed, e.g. by EMPeM? It will be important to understand whether other neutralizing epitopes are targeted. The fact that the anti-Rh5-34EM response peaks after two immunizations and cannot be boosted could be due to the fact that epitope shifts cannot be induced given the size of the immunogen. This might also explain the inability to boost with a third immunization (although it would be interesting to see whether a delayed boost might overcome this).

We agree with the reviewer that these are extremely interesting questions for future studies. As the PfrH5-specific responses plateau with two doses of RH5-34EM, this suggests that antibody responses against this epitope are then saturated, perhaps due to antibodies induced by the first two doses binding to antigen used in the third dose. We speculate about this in the discussion (lines 327-331). It is therefore very likely that boosting with PfrH5 induces synergistic antibodies which bind to a different epitope to that bound by RH5-34EM, thereby improving GIA. EMPeM could be used to help understand this, but is not straight-forward with PfrH5, due to its small size and a shape which appears very similar when rotated by 180°. When we have our final PfrH5-based immunogen, following the strategies outlined in the discussion of this manuscript to get there, we will definitely want to answer the questions raised here by the reviewer.

Are the immunization data representative of independent experiments? An n=1 does not seem to be appropriate.

The experiments in the format presented in the manuscript were only conducted once, with sufficient sample sizes (n=6), thereby minimising use of animals in accordance with our ethics requirements. We treated each animal individually to avoid averaging, therefore giving a true n=6 comparison. The inclusion of three doses of PfrH5 also provided an internal control, allowing comparison of this data set with other published data sets, which also include this same immunogen. We obtained equivalent levels of ELISA reactivity and GIA from three PfrH5 doses in this study as in other studies, giving us confidence that this study was properly conducted and outcomes were comparable with other work. We are confident that this experimental design is appropriate.

While this entire study was conducted once at n=6, we previously conducted a smaller pilot study of three doses of RH5-34EM with three doses of PfrH5 and obtained the same outcome. This provided additional confidence that this n=6 study was robust.

Specific comments:

ELISA: How does the molarity of Rh5 vs Rh5-34EM compare and why was the OD limited to about 1? Could this affect the absorbance-based comparisons.

For both PfrH5 and RH5-34EM, 2µg of antigen was added to each ELISA well. As PfrH5 is ~60kDa, while RH5-34EM is ~15kD, these were not equimolar amounts. All plates contained the same internal controls, which are defined in the methods and development time was

based on the time required for these internal controls to reach ~1.0. This was different for PfRH5 (~15 minutes) and RH5-34EM (~20 minutes). These internal controls allow direct comparison of PfRH5 ELISAs with one another, and RH5-34EM ELISAs with each other, but not PfRH5 ELISAs with RH5-34EM ELISAs. We have taken this into account when drawing conclusions from our data.

Can the authors provide examples of the raw data (similar to the GIA data) as part of the supplement? Were all ELISA experiments that are shown in individual graphs performed in parallel? This seems to be essential given that absorbance data are shown. How many times were the ELISAs repeated?

All of the ELISA data presented in the manuscript (Figure 4cd, Figure 5ab and Figure EV3) is raw and unmanipulated data and raw data is also given in supplementary information. The ELISA measurements were not all performed on the same plates, but we included positive and negative controls, as laid out in the methods and obtained very consistent readings for these controls across all plates. Each ELISA sample was studied once and the consistency obtained for measurements in the six rats in each cohort gives us confidence that this data is reliable and high quality.

Can the authors provide a brief description of how they purified the serum IgG (including the adjustment of concentrations or were these not adjusted?) and how they calibrated the Rh5 concentrations?

This is now included in the methods section

Suppl. Fig. 3: The animals in the RRE group did not respond to the R prime (d27) compared to animals in groups RRR, REE and RER. Why is that? Is it because of the absorbance readout? The inconsistency needs to be addressed.

We agree that it is surprising that the RRE animals did not respond to the PfRH5 prime and mentioned this briefly in lines 242-243. We have added an additional sentence to expand this in lines 243-246, to raise caution about whether this prime dose was appropriately delivered to this cohort by our sub-contractors. This is the one inconsistency in the data set and we agree with the reviewer that it is appropriate to highlight it, such that this one data point can be considered cautiously by readers.

Referee #2 (Remarks for Author):

Summary

The manuscript by Harrison, Alam, Farrell et al. extends the molecular optimization of the PfRH5 antigen, which is the leading blood-stage vaccine candidate for malaria. This work presents a reverse-vaccinology approach, informed by previously reported structure-activity relationships of potent mAbs, as well as for R5.034 which is described in this paper, to immuno-focus the most potent RH5 epitope on a carrier scaffold. The authors use a combination of computational and biophysical approaches to downselect towards a lead antigen candidate, which they evaluate in rodent immunizations with comparisons to full-length RH5. The immunization data presented indicates that the novel engineered RH5 antigen can be multimerized to produce a focused, potentially inhibitory humoral response

towards the target epitope. Importantly, different prime-boost regimens are assessed which suggest that while total sera potency may not be improved by this novel antigen, the quality of the antibodies generated is indeed higher than those induced by the full-length protein. The manuscript is an important contribution to the field of malaria blood-stage vaccine design and would be notably enhanced for publication by addressing the following points.

We thank the reviewer for their positive comments on our work.

Major points

Rodent immunization studies were performed by comparing monomeric full-length RH5 (referred to as RH5.1) with RH5-34EM multimerized on the HBsAg VLP platform. The manuscript requires a more detailed explanation of the immunization doses used in this study, as it is unclear what "equimolar to 2 μ l of soluble RH5.1 protein" means. Additionally, the authors should more clearly indicate that they are comparing monomeric RH5.1 with multimerized RH5-34EM. Would P_fRH5.1-Spy-tag-Spy-catcher-HBsAg not have been a better comparator arm, or perhaps a RH5-34EM monomeric arm in addition to its HBsAg-multimerized state as an important comparator to gauge the impact of multimerization on immunogenicity for this novel immunogen? If this data is not possible, the implications of comparing monomers vs multimers should be stated in the Discussion and these limitations of the study clearly discussed.

We thank the reviewers for noticing the error as instead of 2 μ l we meant 2 μ g. We corrected this and have clarified in the methods section (lines 505-507) what mass of RH5-34EM we calculated to be the same number of moles as 2 μ g of P_fRH5.1.

We agree with the reviewer, that using a VLP-conjugated P_fRH5.1 would be an interesting comparator here. However, when we designed this study, we decided to test our RH5-34EM vaccine against P_fRH5.1 formulated with Matrix-M as this was the leading clinical vaccine candidate at the time. All studies in both humans and Aotus to date have used soluble P_fRH5.1 and conjugation of P_fRH5.1 to VLPs has been extremely challenging, preventing its effective use in the clinic. In contrast, RH5-34EM is readily conjugated to VLPs.

It is true that if we had chosen to use a VLP-conjugated P_fRH5.1 as the comparator, then this might have generated a stronger immune response than soluble P_fRH5.1, but as P_fRH5.1 already generates a much stronger response than RH5-34EM, and as our main conclusions are about the quality of the RH5-34EM-induced immune response, then we are confident that this is not a problem.

At this stage, we will therefore not conduct further rat studies with other versions of P_fRH5.1 or RH5-34EM but instead have, as suggested, discussed the implications of these study design decisions. We have stated the reason for the decision to use soluble P_fRH5.1 in lines 185-188 and highlighted that we used soluble P_fRH5 in the discussion in line 321-323.

It is curious that three doses of RH5.1 will induce intermediate antibody titers against RH5-34EM protein as measured by ELISA (Figure 4c), while two doses of RH5.1 followed by one dose of RH5-34EM produces effectively no RH5-34EM-reactive sera (Figure 5a). It would be helpful for the readers' interpretation of the *in vivo* immunogenicity data for the authors to

expand on whether they believe there is a molecular basis for this finding or if this is due to experimental variability.

We agree with this reviewer that this specific data point (RRE) is out of line with the remaining data. Please see our response to reviewer 1 for how we highlight this in the manuscript.

The immunization data indicates that while the total IgG potency may not have been improved by RH5-34EM, the quality of the RH5-specific antibodies was improved compared to the full-length protein. The conclusion of this manuscript would meaningfully benefit from a discussion section focused on potential strategies to increase the sera titers for RH5-34EM.

We have now included an additional sentence (lines 354-357) into our discussion section on how we might progress rational vaccine design in the light of the findings presented in this manuscript. The likely direction here is to present the immunogen such that exposure of the epitope is increased which exposure of the scaffold is decreased. We have now stated this more clearly.

Lines 203-205: what about EC50? How does the function of the purified IgG compare if this more stringent cut-off is used? This also applies to data reported in Fig. 5. It would be important to present EC50 data perhaps as a Supp Fig. and mention it in the Results, as it also provides a link to the best mAb potency stated in the Introduction.

As already mentioned to reviewer 1, not all of the samples reached EC50, due to lower PfRH5-specific antibody responses to RH5-34EM. Without excluding animals or pooling samples, it is therefore not possible to report EC50 for the full data set but we now include EC50 where possible in Figure EV5.

Minor points

The authors should provide current malaria deaths from the World Malaria Report 2023.

We have updated these numbers.

Could the authors elaborate on the known genetic polymorphisms within the focused epitope selected for RH5-34EM?

None of the residues in RH5-34EM or the epitopes of these tree antibodies, are polymorphic. We have now mentioned this in line 137-138.

It would be informative to the field of research related to antigen scaffolding for the authors to present ELISA data on the scaffold-specific responses induced by RH5-34EM, if available. This would also greatly enhance the understanding of the differences presented for the total IgG titers compared to the RH5-specific IgG titers.

Unfortunately this data is not available. The best way to achieve this would have been to take sera raised through immunisation with RH5-34EM and deplete out PfRH5-specific antibodies before repeating the ELISA. Sadly, this requires more serum than we had available and this was not possible. Our view is that we obtain an approximate answer to this

question simply by comparing Figures 4b and 4c. While it is formally correct to note that the ELISAs for reactivity to PfRH5 and RH5-34EM cannot be quantitatively compared, we do note that they plateau at a similar level. The 10^3 -fold difference in reactivity on the RH5-34EM ELISA when compared with the PfRH5 ELISA when studying the day 70 sera from rats immunised with RH5-34EM therefore is instructive, suggesting that over 99% of antibodies elicited by RH5-34EM are targeted to the scaffold rather than the PfRH5-based epitope. This is why we suggest scaffold masking in the discussion to further improve this immunogen. We have added a few sentences to make this point in lines 253-256.

To better understand the protein engineering workflow and in silico down-selection process, it would be helpful for the authors to provide a supplementary figure detailing the Rosetta pipeline and the predicted energies of the designs.

We have added a new Figure EV1 in which we include a pipeline and a plot of RMSD for the designs vs the Rosetta energy.

The expected protein molecular weights for constructs 1-9 and 3A-C should be indicated next to the SDS-PAGE in Supplementary figure 1a so that the reader can better interpret the migration distance and banding patterns presented on the gel.

The molecular weights for these 12 designs vary little, from 16.2-16.5kDa. We have added this information into the legend for Figure EV1. The pI values vary more, from 5.2-9.1, and might be having a greater effect on gel mobility than molecular weight. Gel mobility is not a great way to determine molecular weight and the gel is provided more so readers can assess sample purity.

Fig 1f: there are large error bars on KD measurements for constructs 3A, 3B and 3C (~50x range in error bars). Could this be improved? If not, can the authors comment on this high variability for this typically precise biophysical measurement?

The error bars in which experiment are obtained from the 95% confidence interval from least squares fit of the equilibrium binding curve as a more high-throughput way to assess binding for a larger number of designs. We then proceeded with the best design to do more accurate kinetic binding measurements. On a log scale, similar confidence intervals (generally ± 1 -2nM) appear larger. Nevertheless, even with these 95% CI values, 3A, 3B and 3C are all still higher affinity than all other designs, only slightly overlapping with 3, but with no others. This therefore supports our conclusion is that these are the most promising designs to take forward and justifies our decision making.

Lines 173-174: the RH5-34EM nomenclature rationale was introduced above (lines 147-148), is it necessary to restate here again?

We have removed this second statement.

Lines 195-196: "It is therefore noteworthy that these two groups showed a similar ELISA reactivity against PfRH5 after three doses, [...]" This statement is misleading. Shouldn't it say: "similar ELISA reactivity against the opposite antigenic probes after three doses"? See Figure 4d that clearly shows the two different antigenic probes used.

We have removed this section of the manuscript, in response to reviewer 3.

The Rfree is quite high (30.46%) for the PfRH5:R5.034 complex despite decent data collection statistics at 2.4 Å resolution. Can the model be further improved, or can the authors provide an explanation for this relatively poor agreement between the model and unbiased X-ray data?

We agree with the reviewer that the PfRH5:R5.034 structure has a higher than usual Rfree, despite extensive refinement, as reflected in the other statistics. In contrast, the RH5-34EM-bound structures both have low Rfree values. We attribute the higher Rfree for PfRH5:RH5-34EM to poorly defined density for the constant domain of the heavy chain of the antibody (chain B). This can be seen in the density and also in the residue-property plots in the validation report. The placement of the constant domain is, nevertheless, clear due to the quality of the density in other regions of the domain. We therefore docked the domain in place, at the expense of Rfree. The density for the variable domains and for the antibody-PfRH5 interface are clear and match well what we observed in the higher resolution, lower Rfree, R5.034:RH5-34EM structure, giving us confidence that our structure is high quality and that our epitope is well defined.

The authors should provide representative maps depicting electron density of antibody-antigen interfaces as a supplemental figure.

Our preference is to not provide a density figure. The PDB and MTZ files have now been released and all readers are free to examine the electron density, which is much better than attempting to interpret from static images, in which the density is shown selectively, at a specific contour level.

The authors should confirm that figure and table callouts are correct as several appear to reference incorrectly. Additionally, the panel orders should match the order of the call outs in the text.

We have checked the callouts.

There are several typos for punctuation and improper spacing between numbers and units throughout the text and figures.

We are confident that these will be put into house style by the copy editing process of the journal.

Referee #3 (Comments on Novelty/Model System for Author):

The experiments were performed in the most logical manner possible.

Referee #3 (Remarks for Author):

Harrison, Alam, Farrell, and colleagues report the design and characterization of an epitope-scaffold immunogen based on the 9AD4/R5.016/R5.034 epitope in PfRH5. The manuscript is clearly written, the results are clearly presented in the figures, and the analysis and interpretation of the data is objective and balanced. I commend the authors on the last point. The manuscript is suitable for publication with only very minor revisions.

We thank the reviewer for their positive assessment of the study.

Major points:

-- Lines 9-11: I hesitate to call this a "major point", but in the absence of major flaws in the manuscript, there is one sentence in the Abstract that should be revised: "In immunised rats, the immunogen induced PfRH5-targeting antibodies that inhibit parasite growth at a thousand-fold lower concentration than those induced through immunisation with PfRH5." That's not quite true as written. The antibodies elicited by RH5-34EM were more potent in the GIA ****per unit PfRH5 binding antibody****. But this is neither surprising nor relevant in real-world vaccination scenarios. The sentence as written implies that the designed immunogen was 1000x more potent, and that simply isn't true. The authors should revise this sentence.

We agree with the reviewer and had not intended to imply that the immunogen was 1000-fold more potent. While our original sentence was correct, and we do think surprising, we agree that this could be misleading without the context of our data. We have therefore rewritten this sentence to "While PfRH5-specific antibodies were induced at lower concentration by the immunogen than by PfRH5, the immunogen induced antibodies that were a thousand-fold more growth inhibitory as a factor of PfRH5-specific antibody concentration."

Minor points:

-- Lines 53-54: "epitope for a subdominant neutralising monoclonal antibody" should be "subdominant epitope for a neutralising monoclonal antibody".

We made this change.

-- Lines 104-107: "We selected design 3, which showed a symmetrical size exclusion chromatography profile, which bound 9AD4 with a high affinity and slow off rate, and which gave the closest predicted root-mean-square-deviation to the starting epitope configuration during the design process" should be "We selected design 3, which showed a symmetrical size exclusion chromatography profile, bound 9AD4 with a high affinity and slow off rate, and gave the closest predicted root-mean-square-deviation to the starting epitope configuration during the design process."

We made this change.

-- Lines 138-142: The conclusion as stated is not supported by the data. "indicate that our epitope mimics... will... induce R5.016 and R5.034" should be "suggest that..." or "indicate that... may induce R5.016- and R5.034-like antibodies."

We made this change.

-- Fig. 2d-f: What is the difference between the blue and red dashed lines? This should be defined in the figure legend.

We made this change.

-- Lines 179-181: But PfrH5.1 was not conjugated to HBsAg? Is that a fair comparison? The authors should clarify this point and justify comparing a particulate to a non-particulate immunogen.

Please note our response to reviewer 1 in which we respond to this point.

-- Fig. 4b-d: Is this really Absorbance on the y axis, or is it a reciprocal titer (i.e., ED50)?

The y axes on the ELISA graphs are absorbance (at 405nm) and show the raw data.

-- Lines 195-198: You cannot quantitatively compare ELISA titers against two different antigens laid down on the ELISA plate. Binding to the plate, and therefore the apparent titer, can vary widely for different antigens. As a result, the authors' claim here is invalid. If the authors wish to support this claim, they should determine the absolute concentration of antibody specific to each antigen by using a standard curve in their ELISAs. Otherwise, the authors should remove this claim.

We accept this point from the reviewer. We had pointed out as an interesting observation, rather than designing an experiment to robustly test this point. On reflection, we have removed this comparison and the figure panel from the manuscript. The data remains in Figures 4b and 4c without the comparison being highlighted.

-- Line 270: This is "Reverse vaccinology 2.0" (Rappuoli), or "Structure-based antigen design". The authors should consider citing Rappuoli or Burton's 2002 classic, "Antibodies, viruses, and vaccines", at the end of this sentence.

We made this change and cited Burton 2002

-- Line 298: "effective" should be "effectively".

We made this change

13th Aug 2024

Dear Dr. Higgins,

Thank you for the submission of your revised manuscript to EMBO Molecular Medicine and please accept my apologies for the delay in getting back to you due to the holiday season. I am pleased to inform you that we will be able to accept your manuscript pending the following final amendments:

- 1) Please implement all referee #1 suggestions.
- 2) In the main manuscript file, please do the following:
 - Please address all comments suggested by our data editors listed below:
 - o Data availability section:
 1. Please note that the specific URLs for 8RZ0, 8RZ1 and 8RZ2 datasets are not provided in the data availability statement.
 - o Figure legends:
 1. Please define the annotated p values ****/*** as well as provide the exact p-values for the same in the legend of figure EV 5; as appropriate.
 2. Please note that the exact p values are not provided in the legends of figures 4g; 5c.
 3. Please indicate the statistical test used for data analysis in the legends of figures EV 5.
 4. Please note that information related to n is missing in the legends of figures 4b-c, e.
 5. Please note that n=2 in figure 1f.
 6. Please note that the measure of center for the error bars needs to be defined in the legend of figure 1f.
 - The manuscript sections should be in the following order: Title page - Abstract & Keywords - Introduction - Results - Discussion - Methods - Data Availability - Acknowledgments - Disclosure Statement & Competing Interests - References - Figure Legends - (Main Tables with legends) - Expanded View Figure Legends.
 - Add up to 5 keywords.
 - Add callouts for Figure 3B and 3C.
 - Rename "Competing interests" to "Disclosure and competing interests statement". We updated our journal's competing interests policy in January 2022 and request authors to consider both actual and perceived competing interests. Please review the policy <https://www.embopress.org/competing-interests> and update your competing interests if necessary.
 - Author contributions: Please remove it from the manuscript and specify author contributions in our submission system. CRediT has replaced the traditional author contributions section because it offers a systematic machine-readable author contributions format that allows for more effective research assessment. You are encouraged to use the free text boxes beneath each contributing author's name to add specific details on the author's contribution. More information is available in our guide to authors:
<https://www.embopress.org/page/journal/17574684/authorguide#authorshipguidelines>
 - Please include structured Methods section that includes a Reagents and Tools Table followed by a Methods and Protocols section. More information on how to adhere to this format as well as downloadable templates (.docx) for the Reagents and Tools Table can be found in our author guidelines: <https://www.embopress.org/page/journal/17574684/authorguide#structuredmethods>
An example of a paper with Structured Methods can be found here:
<https://www.embopress.org/doi/full/10.1038/s44320-024-00037-6#sec-4>
 - Indicate in legends number and nature of replicates and exact p= values, not a range, along with the statistical test used. To keep the figures "clear" some authors found providing an Appendix table Sx with all exact p-values preferable. You are welcome to do this if you want to.
 - In data availability statement please add the specific URLs for all deposited datasets.
 - Correct the reference citation in the text and reference list. In the text a reference should be cited by author and year of publication. Include a space between a word and the opening parenthesis of the reference that follows. In the reference list, citations should be listed in alphabetical order. Where there are more than 10 authors on a paper, 10 will be listed, followed by "et al.". Also, please remove DOIs. Please check "Author Guidelines" for more information.
 - <https://www.embopress.org/page/journal/17574684/authorguide#referencesformat>
- 3) Tables: Please remove all EV tables from the main manuscript and upload them as separate .doc files. Also, please remove colors from Table EV1 and indicate interactions with e.g. + and -.
- 4) Funding: Please make sure that information about all sources of funding are complete in both our submission system and in the manuscript.
- 5) The Paper Explained: Please add it to the main manuscript file.
- 6) Synopsis: Please check your synopsis text and image before submission with your revised manuscript. Please be aware that in the proof stage minor corrections only are allowed (e.g., typos).
- 7) As part of the EMBO Publications transparent editorial process initiative (see our Editorial at <http://embomolmed.embopress.org/content/2/9/329>), EMBO Molecular Medicine will publish online a Review Process File (RPF) to accompany accepted manuscripts. This file will be published in conjunction with your paper and will include the anonymous referee reports, your point-by-point response and all pertinent correspondence relating to the manuscript. Let us know whether you agree with the publication of the RPF and as here, if you want to remove or not any figures from it prior to publication. Please note that the Authors checklist will be published at the end of the RPF.
- 8) Please provide a point-by-point letter INCLUDING my comments as well as the reviewer's reports and your detailed

responses (as Word file).

I look forward to reading a new revised version of your manuscript as soon as possible.

Yours sincerely,

Zeljko Durdevic

*** Instructions to submit your revised manuscript ***

- 1) a .docx formatted version of the manuscript text (including Figure legends and tables)
- 2) Separate figure files*
- 3) supplemental information as Expanded View and/or Appendix. Please carefully check the authors guidelines for formatting Expanded view and Appendix figures and tables at <https://www.embopress.org/page/journal/17574684/authorguide#expandedview>
- 4) a letter INCLUDING the reviewer's reports and your detailed responses to their comments (as Word file).
- 5) The paper explained: EMBO Molecular Medicine articles are accompanied by a summary of the articles to emphasize the major findings in the paper and their medical implications for the non-specialist reader. Please provide a draft summary of your article highlighting
 - the medical issue you are addressing,
 - the results obtained and
 - their clinical impact.This may be edited to ensure that readers understand the significance and context of the research. Please refer to any of our published articles for an example.
- 6) Author contributions: the contribution of every author must be detailed in a separate section.
- 7) EMBO Molecular Medicine now requires a complete author checklist (<https://www.embopress.org/page/journal/17574684/authorguide>) to be submitted with all revised manuscripts. Please use the checklist as guideline for the sort of information we need WITHIN the manuscript. The checklist should only be filled with page numbers where the information can be found. This is particularly important for animal reporting, antibody dilutions (missing) and exact values and n that should be indicated instead of a range.
- 8) Every published paper now includes a 'Synopsis' to further enhance discoverability. Synopses are displayed on the journal webpage and are freely accessible to all readers. They include a short stand first (maximum of 300 characters, including space)

as well as 2-5 one sentence bullet points that summarise the paper. Please write the bullet points to summarise the key NEW findings. They should be designed to be complementary to the abstract - i.e. not repeat the same text. We encourage inclusion of key acronyms and quantitative information (maximum of 30 words / bullet point). Please use the passive voice. Please attach these in a separate file or send them by email, we will incorporate them accordingly.

You are also welcome to suggest a striking image or visual abstract to illustrate your article. If you do please provide a jpeg file 550 px-wide x 300-600px high.

9) A Conflict of Interest statement should be provided in the main text

10) Please note that we now mandate that all corresponding authors list an ORCID digital identifier. This takes <90 seconds to complete. We encourage all authors to supply an ORCID identifier, which will be linked to their name for unambiguous name identification.

Currently, our records indicate that the ORCID for your account is 0000-0002-2870-1955.

Link Not Available

11) Include a Reagents and Tools Table as part of the Methods section, which can be downloaded from our author guidelines (<https://www.embopress.org/page/journal/17574684/authorguide#structuredmethods>)

Photos 400-800 DPI

*Additional important information regarding figures and illustrations can be found at

<https://bit.ly/EMBOPressFigurePreparationGuideline>. See also figure legend preparation guidelines:

<https://www.embopress.org/page/journal/17574684/authorguide#figureformat>

***** Reviewer's comments *****

Referee #1 (Comments on Novelty/Model System for Author):

The authors have addressed my concerns in the rebuttal letter. Some modifications were also made to the manuscript. It would be wonderful if the authors could discuss the other points in a "limitations to the study" section. For example, the GIA assay seems to be a major limiting factor, which requires purified IgG at high concentration. The development of a more sensitive assay is probably needed to advance the field. The n=1 in vivo study should also be mentioned/discussed.

Referee #1 (Remarks for Author):

This is an important study for the field.

Referee #2 (Remarks for Author):

The authors have addressed all concerns raised and provide a revised manuscript suitable for publication.

The authors addressed the remaining editorial issues.

20th Aug 2024

Dear Dr. Higgins,

We are pleased to inform you that your manuscript is accepted for publication and is now being sent to our publisher to be included in the next available issue of EMBO Molecular Medicine.
